# Locus coeruleus-CA1 projections are involved in chronic depressive stress-induced hippocampal vulnerability to transient global ischaemia

Qian Zhang[1,6], Dian Xing Hu[2,6], Feng He[1], Chun Yang Li[2], Guang Jian Qi[1], Hong Wei Cai[1], Tong Xia Li[1], Jie Ming[3], Pei Zhang[1,4,5], Xiao Qian Chen[2,4,5] & Bo Tian[1,4,5]

Depression and transient ischaemic attack represent the common psychological and neurological diseases, respectively, and are tightly associated. However, studies of depression-affected ischaemic attack have been limited to epidemiological evidences, and the neural circuits underlying depression-modulated ischaemic injury remain unknown. Here, we find that chronic social defeat stress (CSDS) and chronic footshock stress (CFS) exacerbate CA1 neuron loss and spatial learning/memory impairment after a short transient global ischaemia (TGI) attack in mice. Whole-brain mapping of direct outputs of locus coeruleus (LC)-tyrosine hydroxylase (TH, Th:) positive neurons reveals that LC-CA1 projections are decreased in CSDS or CFS mice. Furthermore, using designer receptors exclusively activated by designer drugs (DREADDs)-based chemogenetic tools, we determine that Th:LC-CA1 circuit is necessary and sufficient for depression-induced aggravated outcomes of TGI. Collectively, we suggest that Th:LC-CA1 pathway plays a crucial role in depression-induced TGI vulnerability and offers a potential intervention for preventing depression-related transient ischaemic attack.

[1] Department of Neurobiology, School of Basic Medicine, Tongji Medical College, Huazhong University of Science and Technology, Wuhan 430030, China. [2] Department of Pathophysiology, School of Basic Medicine, Tongji Medical College, Huazhong University of Science and Technology, Wuhan 430030, China. [3] Department of Breast and Thyroid Surgery, Union Hospital, Huazhong University of Science and Technology, Wuhan 430022, China. [4] Institute for Brain Research, Huazhong University of Science and Technology, Wuhan 430030, China. [5] Key Laboratory of Neurological Diseases, Ministry of Education, Wuhan 430030, China. [6] These authors contributed equally: Qian Zhang, Dian Xing Hu. Correspondence and requests for materials should be addressed to P.Z. (email: zhangpei@hust.edu.cn) or to X.Q.C. (email: chenxq@mails.tjmu.edu.cn) or to B.T. (email: tianbo@mails.tjmu.edu.cn)

Transient global ischaemia (TGI) is characterised as a brief neurological dysfunction episode caused by a loss of blood flow (ischaemia) in the brain, spinal cord or retina[1]. Global brain ischaemia results in delayed cell death in vulnerable neuron populations, including hippocampal CA1 pyramidal neurons and is associated with impaired cognition and memory defects in both humans and animals[2–4]. Studies have implicated various processes, such as calcium influx, glutamate neurotoxicity, cellular suicide gene expression, apoptotic protein activation, endoplasmic reticulum dysfunction and mitochondrial dysfunction that may contribute to this vulnerability to neuronal death and spatial memory dysfunction. Recently, numerous studies have focussed on understanding and manipulating neural circuits to prevent ischaemia-related insults. This is because neural circuit activities can be regulated by neuromodulators to produce various outputs to modulate behaviour as part of a rational strategic reallocation of resources to vital functions[5]. Therefore, understanding the related neural circuits underlying this ubiquitous neuromodulatory connectivity and how they influence TGI are critical steps for improving patient recovery from ischaemic stroke.

Epidemiological studies have demonstrated that depression and TGI are highly correlated. Depression is a frequent and recurrent mood disorder affecting daily functioning and quality of life and increases morbidity and mortality[6]. For decades, post-stroke depression has been considered the most frequent and important neuropsychiatric consequence of stroke[7]. Unexpectedly, recent meta-analyses of cohort studies have suggested that depression is associated with a 34–63% excess risk of all strokes combined[8–10]. Studies have suggested several pathways through which depression or depressive symptoms may influence stroke[11]. Hypothesised mechanisms underlying this process include accumulation of biological damage, such as in atherosclerosis, hypertension, cerebrovascular reactivity and atrial fibrillation[12–14], which increases the risk of experiencing stroke events. We aim to elucidate the mechanisms underlying depression conferring TGI susceptibility.

The locus coeruleus (LC) is the major source of norepinephrine (NE) in the brain and has been linked to the pathogenesis of depression[15]. The LC sends numerous outputs to numerous regions of the forebrain and spinal cord to modulate a diverse range of physiological functions, including cognition, arousal, memory and pain processing[16–18]. It is well established that the LC activity in response to chronic stress leads to a deficit in noradrenergic neuron activity and a robust decrease in the release of NE[19–21]. The LC system is a main component of the centrally induced fight-or-flight response associated with numerous environmental stressors, including predator and social stress, which activate the LC system[22–24]. In addition, the noradrenergic projections from the LC to the basolateral amygdala (BLA) mediate anxiety-like behaviour in an adrenergic receptor-dependent manner[25]. Recent studies have begun to illustrate how the complex efferent system of LC neurons selectively mediates specific behaviours. Moreover, LC-TH$^+$ neurons project more profusely to hippocampus subregions, such as CA1 and CA3, involved in certain mouse behaviours related to environmental novelty than ventral tegmental area (VTA) TH$^+$ neurons[26]. In addition, dopaminergic axons and the subsequent release of dopamine in the dorsal hippocampus originate from LC neurons, as opposed to those of the VTA, to modulate spatial learning and memory[27]. Indeed, the activation of LC-TH$^+$ projections has been shown to induce an accelerated defensive response to looming[28]. Given that TGI is a major cause of chronic neurological disabilities, including spatial memory impairment and delayed neuronal death of CA1 region neurons within the hippocampus[29], it is possible that the connections between depression and TGI may be targeted and manipulated through the Th:LC-CA1 neural circuit.

In this study, we determine that depression in mice, including chronic social defeat stress (CSDS) and chronic footshock stress (CFS), induce an aggravated response to TGI that is found to be mediated by the Th:LC-CA1 pathway. In particular, the projection strength of the Th:LC-CA1 circuit is significantly reduced in mice susceptible to CSDS and CFS, as evidenced by fewer axons in the CA1 region compared with control mice. In addition, we demonstrate that selective chemogenetic inhibition of TH$^+$ projections from the LC to the CA1 served to mimic depression-like behaviours and induce a worsening response to TGI, further confirming the necessity of this pathway. Furthermore, the chemogenetic activation of the Th:LC-CA1 pathway reverses aggravated TGI responses after CSDS and CFS. Collectively, our findings reveal a previously undefined role for Th:LC-CA1 projections in how depression increases TGI vulnerability.

## Results

**Time-dependent effects of TGI on hippocampal function.** To clarify whether depression increased the vulnerability of the hippocampus to ischaemic injury or not, we first needed to establish a reliable ischaemic model that specifically affects hippocampal function. Among the various ischaemic models evaluated[30,31], the TGI-based model is the most reproducible. In this model, spatial memory impairment and hippocampal CA1 pyramidal neuron death is selectively induced, thus mimicking ischaemic-related chronic neurological disabilities[3,32,33]. To identify the appropriate time-point of TGI that induces specific CA1 neuronal death and memory loss, mouse brains were subjected to consecutive time-points (10, 20 and 30 min) of transient ischaemia (bilateral common carotid artery occlusion) followed by full reperfusion. A battery of behavioural tests (open field test, OFT; grip strength test, GST; rotarod test, RT; and Kondziela's inverted screen test, KIST) were then performed at Days 7 post recovery, followed by a spatial learning and memory test of Morris water maze (MWM) (Fig. 1a). Our results demonstrated that TGI treatment within 30 min did not alter animal motor locomotion (via the OFT, Fig. 1b), maximal skeletal muscle strength (via GST, Fig. 1c) or motor coordination (via the RT and KIST, Fig. 1d, e). However, mice with the same TGI time-points exhibited significant differences in the MWM test, which is highly sensitive to hippocampal damage and reflects performance related to spatial learning and memory[34]. Figure 1f–i showed that the spatial learning ability of the mice was not altered after TGI-10 min, but was significantly impaired after TGI-20 min and TGI-30 min compared with Sham-operated mice in a 6-day MWM training session. After the training session, the platform was removed and mice were allowed to search for the missing platform in the target quadrant in a MWM probe test that reflects spatial memory performance. Figure 1j showed the representative tracing path for each group in the probe trial. Statistical analysis of the results demonstrated that TGI-10 min did not alter the mouse target-searching time compared with sham treatment, however, TGI-20 min and TGI-30 min mice had significantly decreased mouse target-searching time (Fig. 1k), while the total searching distance did not differ among the groups (Fig. 1l). These results identified 10–20 min of TGI as the threshold time-period for inducing learning and memory impairment in mice.

After the last probe trial, mice were euthanised and brain sections were stained with NeuN to detect surviving neurons[35]. Figure 1m showed the representative NeuN fluorescent staining of the hippocampus for each group, focussing on the survival of CA1 pyramidal neurons, which are associated with learning and memory[36–38]. Our results demonstrated that numbers of NeuN$^+$-pyramidal neurons in the CA1 region were significantly reduced after TGI-20 min and TGI-30 min, but not after TGI-10

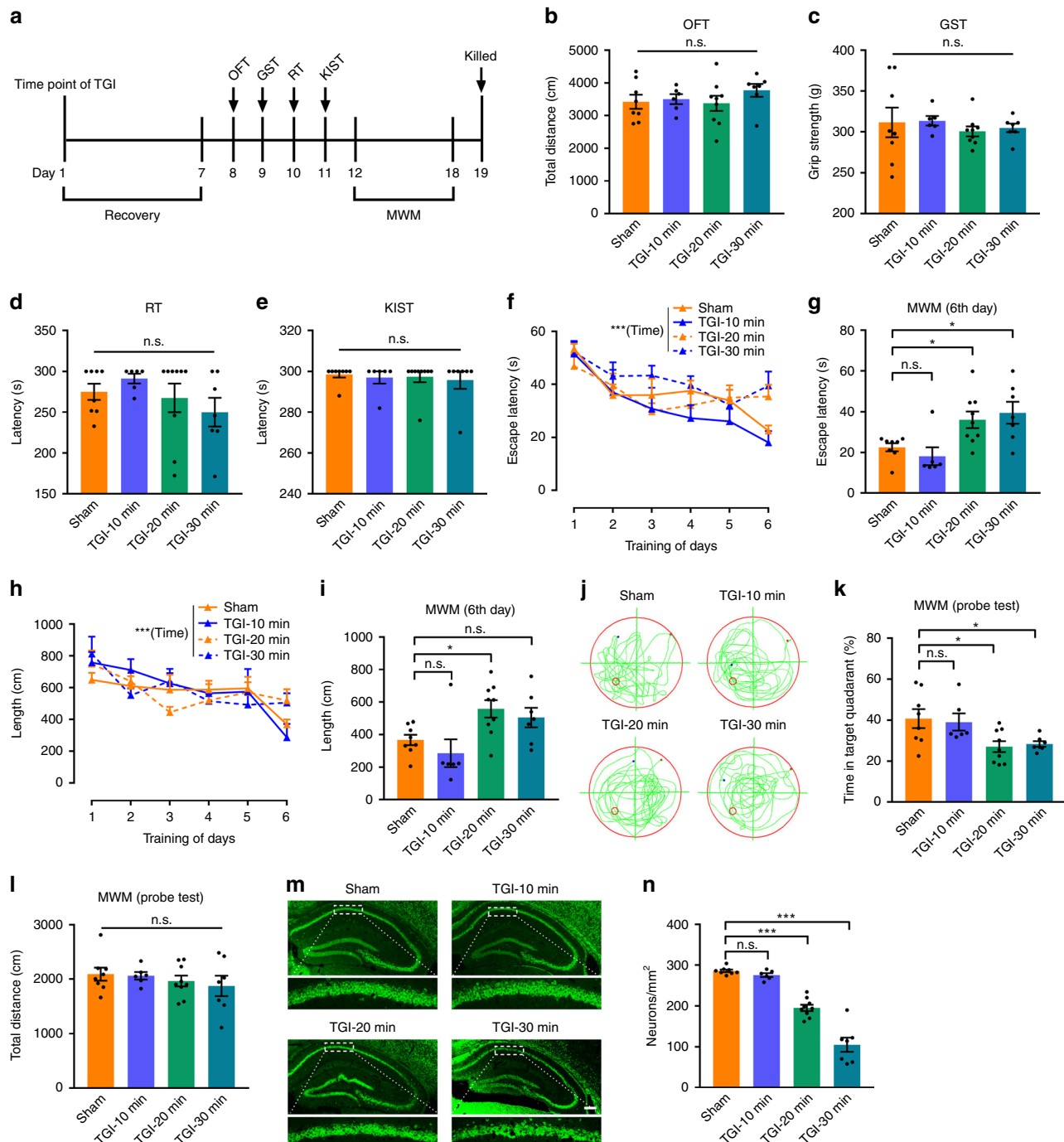

**Fig. 1** Time-dependent hippocampal damage after transient global ischaemia. **a** Experimental scheme. Mice were exposed to various time-points of TGI at 10, 20 and 30 min and then evaluated by open field test (OFT), grip strength test (GST), rotarod test (RT), Kondziela's inverted screen test (KIST) and Morris water maze (MWM). Measurements for **b** locomotor activity were performed by OFT (one-way ANOVA, P = 0.5604), **c** muscular strength by the GST (one-way ANOVA, $P = 0.8167$) and **d** motor function by RT (one-way ANOVA, P = 0.3309) and **e** KIST (one-way ANOVA, P = 0.9277). **f** Mean latencies to a hidden platform from the acquisition trials (RM two-way ANOVA, interaction: $P = 0.4005$, time: $P < 0.0001***$, treatment: $P = 0.0691$). **g** The latency to reach a hidden platform during the training tests on day 6 (one-way ANOVA, $P = 0.0047**$; post hoc Dunnett's test, Sham vs. TGI-10 min, $P = 0.8035$, Sham vs. TGI-20 min, $P = 0.0493*$, Sham vs. TGI-30 min, $P = 0.0181*$). **h** The mean distance to reach a hidden platform (RM two-way ANOVA, interaction: $P = 0.2621$, time: $P < 0.0001***$, treatment: $P = 0.9592$). **i** The swim length to reach a hidden platform on day 6 (one-way ANOVA, $P = 0.0141*$; post hoc Dunnett's test, Sham vs. TGI-10 min, $P = 0.6527$, Sham vs. TGI-20 min, $P = 0.0485*$, Sham vs. TGI-30 min, $P = 0.2347$). **j** Representative path tracings from the probe trials on day 7. **k** The percentage of time spent in the target quadrant during the probe trial (one-way ANOVA, $P = 0.0125*$; post hoc Dunnett's test, Sham vs. TGI-10 min, $P = 0.9758$, Sham vs. TGI-20 min, $P = 0.0159*$, Sham vs. TGI-30 min, $P = 0.0446*$). **l** The total distance from the probe trials on day 7 (one-way ANOVA, $P = 0.6342$). **m** Representative images of the hippocampus (top) and CA1 area (bottom). The slices were stained with anti-NeuN (Green). Scaling bar: 250 μm (top), 50 μm (bottom). **n** Quantification of CA1 neurons by immunofluorescence as shown in (**m**), one-way ANOVA, $P < 0.0001***$, post hoc Dunnett's test (Sham vs. TGI-10 min, $P = 0.8109$, Sham vs. TGI-20 min, $P < 0.0001***$, Sham vs. TGI-30 min, $P < 0.0001***$). Sham ($n = 8$), TGI-10 min ($n = 6$), TGI-20 min ($n = 9$), TGI-30 min ($n = 7$). All data are displayed as means ± s.e.m

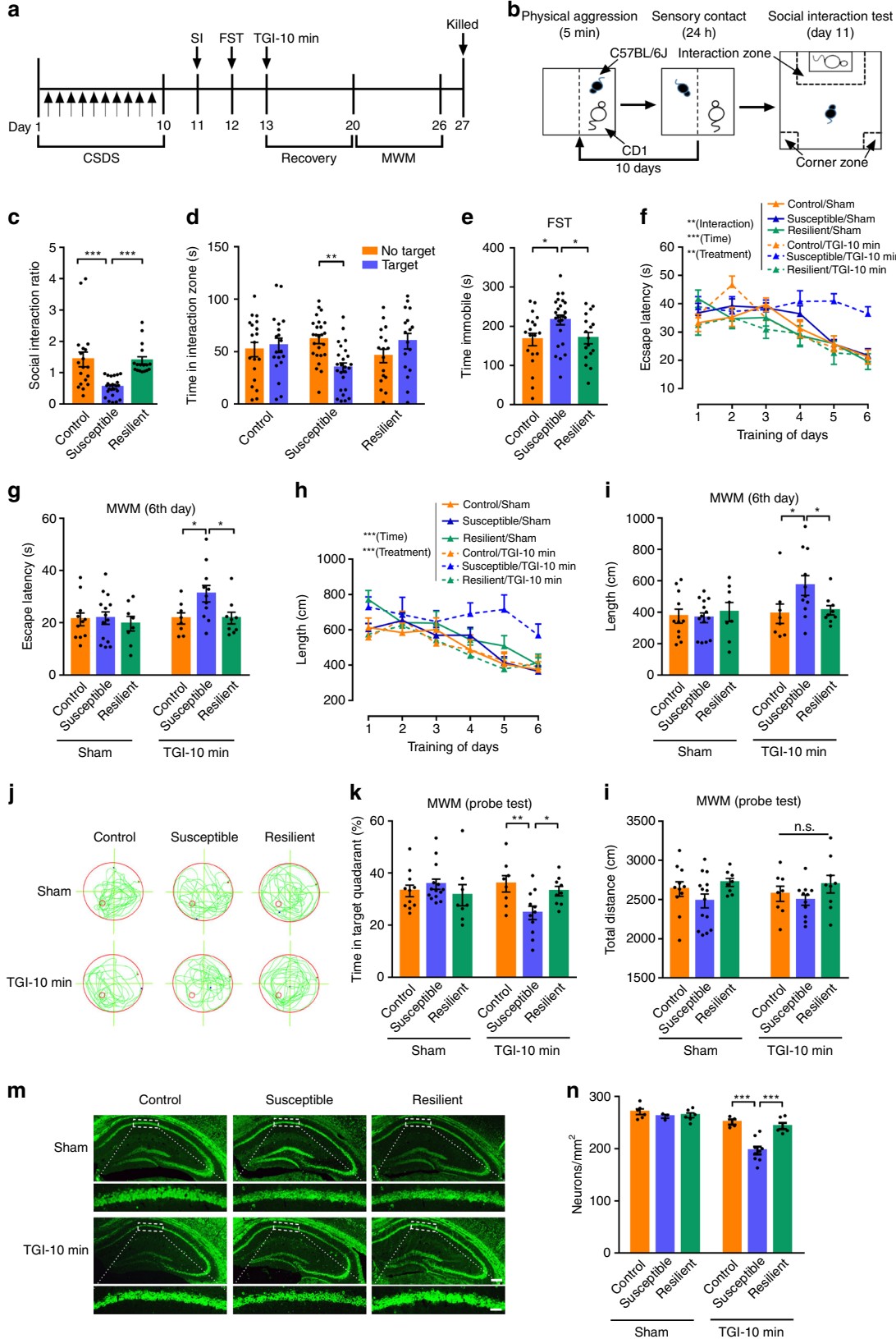

min, compared with sham mice (Fig. 1m, n). Taken together, the above experimental results show that TGI-20 min and TGI-30 min could cause the impairment of learning (at the later stage of training session) and memory, and also loss of the CA1 neurons, while TGI-10 min had no obvious damage to mice. Thus, we chose TGI-10 min in the following experiments in order to observe the effects of depression on TGI-related learning and memory impairment.

**Fig. 2** Chronic social defeat stress aggravates TGI-induced hippocampal injury. **a** Experimental timeline. **b** Behavioural paradigm of repeated chronic social defeat stress (CSDS) experiments. **c** The social interaction ratio by social interaction (SI) test (one-way ANOVA, $P < 0.0001$***; post hoc Dunnett's test, Control vs. Susceptible, $P < 0.0001$***, Susceptible vs. Resilient, $P = 0.0002$***). Control ($n = 19$), Susceptible ($n = 25$), Resilient ($n = 17$). **d** The time in the interaction zone (two-way ANOVA, interaction: $P = 0.0011$**, phenotype: $P = 0.5481$, No target/Target: $P = 0.5378$; post hoc Dunnett's test, Control/No target vs. Control/Target, $P = 0.9542$, Susceptible/No target vs. Susceptible/Target, $P = 0.0012$**, Resilient/No target vs. Resilient/Target, $P = 0.3180$). **e** The immobility time of forced swimming test (FST) (one-way ANOVA, $P = 0.0216$*; post hoc Dunnett's test, Control vs. Susceptible, $P = 0.0253$*, Susceptible vs. Resilient, $P = 0.0474$*). **f** Spatial learning curves in MWM test (RM two-way ANOVA, interaction: $P = 0.0091$**, time: $P < 0.0001$***, treatment: $P = 0.0063$**). Control/Sham ($n = 11$), Control/TGI ($n = 8$), Susceptible/Sham ($n = 14$), Susceptible/TGI ($n = 11$), Resilient/Sham ($n = 8$) and Resilient/TGI ($n = 9$). **g** The latency on day 6 (two-way ANOVA, interaction: $P = 0.1809$, phenotype: $P = 0.0581$, treatment: $P = 0.076$; post hoc Dunnett's test, Control/TGI-10 min vs. Susceptible/TGI-10 min, $P = 0.0336$*, Susceptible/TGI-10 min vs. Resilient/TGI-10 min, $P = 0.0303$*). **h** Swimming distance during training session (RM two-way ANOVA, interaction: $P = 0.6481$, time: $P < 0.0001$***, treatment: $P = 0.0004$***). **i** The swimming length on day 6 (two-way ANOVA, interaction: $P = 0.0688$, phenotype: $P = 0.1882$, treatment: $P = 0.0619$; post hoc Dunnett's test, Control/TGI-10 min vs. Susceptible/TGI-10 min, $P = 0.0293$*, Susceptible/TGI-10 min vs. Resilient/TGI-10 min, $P = 0.0498$*). **j** Representative swimming traces on day 7. **k** The percentage of time spent in the target quadrant (two-way ANOVA, interaction: $P = 0.0159$*, phenotype: $P = 0.2483$, treatment: $P = 0.2639$; post hoc Dunnett's test, Control/TGI-10 min vs. Susceptible/TGI-10 min, $P = 0.0095$**, Susceptible/TGI-10 min vs. Resilient/TGI-10 min, $P = 0.0499$*). **l** Total distance during the probe test (two-way ANOVA, interaction: $P = 0.9241$, phenotype: $P = 0.0542$, treatment: $P = 0.7465$). **m** Representative fluorescent images of hippocampus (top, scaling bar = 250 μm) and a higher magnification CA1 area (bottom, scaling bar = 50 μm). **n** Quantification of surviving CA1 pyramidal neurons. Control/Sham ($n = 6$), Control/TGI-10 min ($n = 4$), Susceptible/Sham ($n = 3$), Susceptible /TGI-10 min ($n = 9$), Resilient/Sham ($n = 6$), Resilient/TGI-10 min ($n = 6$). Two-way ANOVA, interaction: $P = 0.0045$**, phenotype: $P = 0.0004$***, treatment: $P < 0.0001$***; post hoc Dunnett's test, Control/TGI-10 min vs. Susceptible/TGI-10 min, $P < 0.0001$***, Susceptible/TGI-10 min vs. Resilient/TGI-10 min, $P < 0.0001$***. All data are displayed as means ± s.e.m.

**CSDS aggravates TGI-induced hippocampal damage**. To illuminate the potential effects of depression on TGI, mice were exposed to a consecutive 10-day CSDS treatment schedule (Fig. 2a, b). CSDS has been shown to reliably induce an array of depressive-like phenotypes in mice that parallel those seen in humans[39]. In addition, CSDS mouse models display robust predictive validity, with chronic (but not acute) antidepressant treatment reversing its behavioural effects[40]. After 10 days of CSDS, mice could be separated into resilient and susceptible groups, based on social avoidance testing. As expected, susceptible mice showed markedly decreased social interaction (SI) in the presence of CD1 mice, whereas resilient individuals exhibited SI ratio similar to that of undefeated control mice[41] (Fig. 2c, d). Compared with non-defeated controls and resilient mice, the susceptible group displayed a significant increase in the percentage of immobile time during the forced swimming test (FST) (Fig. 2e).

Next, we validated whether mice subjected to CSDS treatment displayed an aggravated vulnerability to spatial learning and memory impairment (MWM test) and CA1 neuronal death after TGI or not. First, we assessed learning and memory in control, resilient and susceptible mice subjected to TGI-10 min or sham operation by using the MWM test. Remarkably, only Susceptible/TGI-10 min mice demonstrated poor learning performances, while the other groups (Control/Sham, Susceptible/Sham, Resilient/Sham, Control/TGI-10 min and Resilient/TGI-10 min) showed normal learning behaviours (Fig. 2f–i). Similarly, in the probe trial, Susceptible/TGI mice represented the only group that did not remember the platform location and did not exhibit a persistent memory for the target quadrant (Fig. 2j, k). To exclude the possibility that this decrease in spatial learning and memory resulted from the change in locomotion, we compared the swimming lengths of mice during the memory probe test on day 7 by removing the platform. As we expected, the swimming length was not significantly altered based on motor activity (Fig. 2l). To evaluate the functional basis of the CSDS-enhanced memory loss observed in TGI mice at the cellular level, hippocampal sections were stained with NeuN to detect CA1 surviving neurons[35]. In accordance with the MWM test, only Susceptible/TGI-10 min mice showed severe CA1 pyramidal neuron losses

(Fig. 2m, n). Taken together, these results indicated that depression may be a critical risk factor in determining TGI vulnerability.

**CFS confers TGI susceptibility to hippocampal injury**. Our findings that CSDS aggravated TGI-induced spatial memory loss support our initial speculation, i.e., depression is an important pathological condition for transient ischaemic attack. To further test this hypothesis, a CFS mouse model of depression was used (Fig. 3a). Mice were subjected to a sequential 14-day footshock procedure, which resulted in evident depression-like behaviour as indicated by a robust increase in immobile time in the FST (Fig. 3b). Statistical analysis of the results demonstrated that the latency to find the platform and the swim length in the MWM test were significantly longer in the CFS/TGI-10 min mice compared with the Control/TGI-10 min group (Fig. 3c–f). These results further confirmed that CFS impaired spatial learning ability in TGI mice. Furthermore, in the probe trial, the CFS/TGI-10 min group spent less time in the target quadrant compared with the Control/TGI-10 min group (Fig. 3g, h), which demonstrates impaired spatial memory in the CFS/TGI-10 min mice. For all groups, no differences in locomotor activity were observed as evaluated by the swimming length (Fig. 3i). Consistently, NeuN fluorescent immunostaining revealed substantial neuronal loss in the CA1 regions of the CFS/TGI-10 min group compared with the other groups (Fig. 3j, k). These data clearly demonstrated that CFS aggravated TGI-induced spatial memory impairment and CA1 neuronal death, which is consistent with the CSDS-stress response in TGI mice.

**Mapping direct outputs of LC-TH$^+$ neurons in depressive mice**. Numerous studies have demonstrated that the efferent projections of the LC are broad, while the TH$^+$ axonal projections of the LC in depression remain largely unknown. To address this issue, we utilised a cell type-specific tracing technique, as well as a viral genetic strategy that permitted the brain-wide mapping of direct output projections in defined neuronal populations[42,43]. Briefly, we unilaterally and stereotactically injected a mixture of two viruses (AAV-TH-Cre and AAV-DIO-ChR2-EYFP) into the LC region (Fig. 4a, b), which resulted in the expression of EYFP

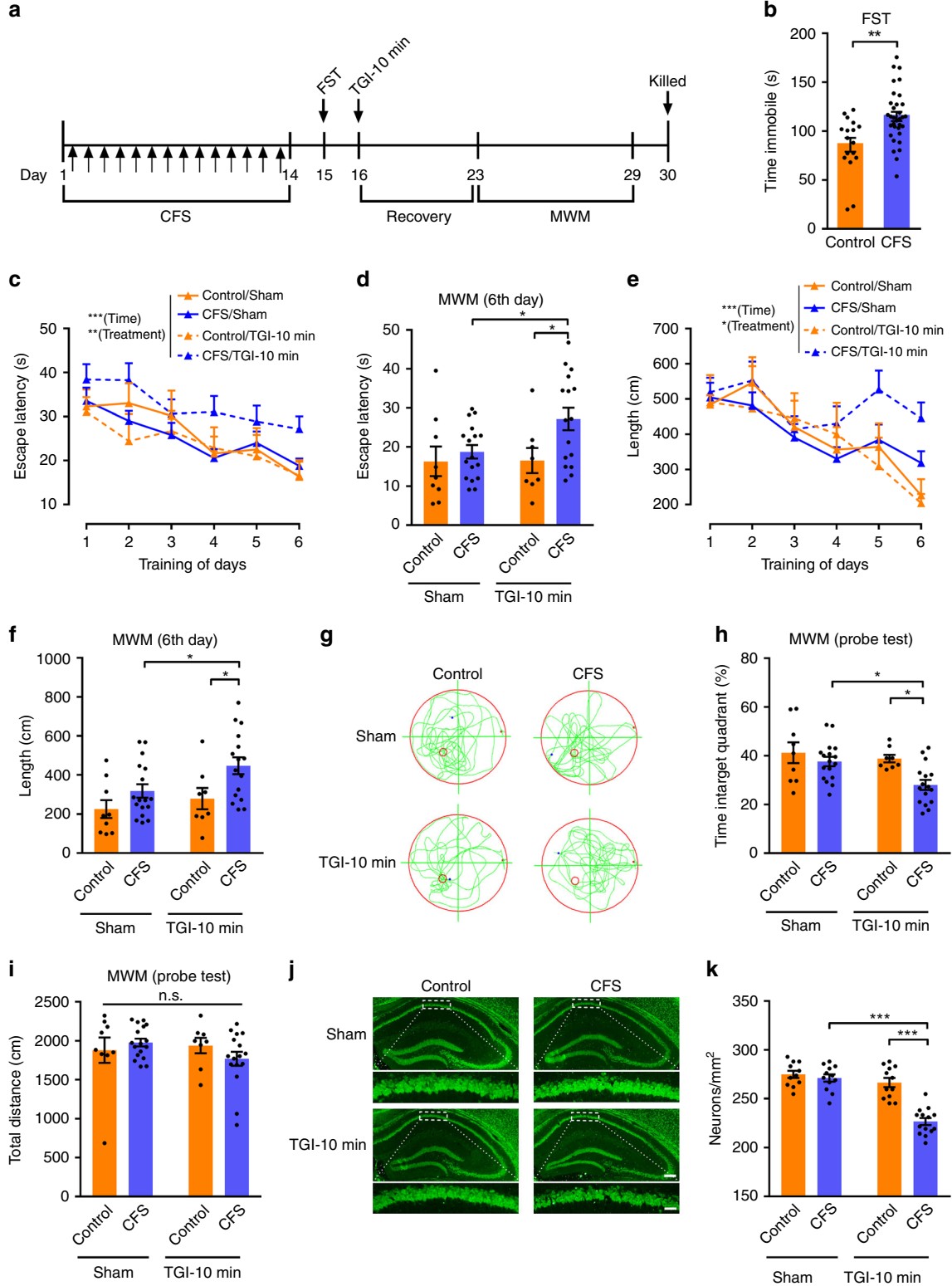

fluorescence in the brain-wide axonal arbours of each TH[+] neuron population. After a 4-week recovery period, mice were subjected to a continuous 10 days of CSDS (Fig. 4c). Next, we quantified the EYFP-positive axonal output patterns of the area population throughout the brain. The series of coronal sections for LC-TH[+] neurons in targeted cases and the average normalised fraction of EYFP[+] axons for each brain region are displayed in Fig. 4d, e, respectively. We found a very sparse expression of LC-TH[+] axons in the CA1 region of the dorsal hippocampus in susceptible mice compared with that in the control and resilient groups (Fig. 4f–i). Since LC-derived TH staining in LHA, ML and SNc was diminished in resilient mice after CSDS, we counted the number of neurons via NeuN-staining in these regions in CSDS-susceptible mice after TGI-10 min (Supplementary Fig. 1a). The

**Fig. 3** Chronic footshock stress confers TGI susceptibility to hippocampal damage. **a** Overall schematic of the methodology utilised. **b** Immobility time was measured by FST (unpaired two-tailed Student's $t$ test, $P = 0.0011^{**}$). Control ($n = 17$), CFS ($n = 33$). **c** The escape latency of water maze (RM two-way ANOVA, interaction: $P = 0.9827$, time: $P < 0.0001^{***}$, treatment: $P = 0.0027^{**}$). Control/Sham ($n = 9$), Control/TGI ($n = 8$), CFS/Sham ($n = 17$), CFS/TGI ($n = 16$). **d** The latency on day 6 (two-way ANOVA, interaction: $P = 0.1692$, Control vs. CFS: $P = 0.03^*$, Sham vs. TGI-10 min: $P = 0.1469$; post hoc Dunnett's test, Control/TGI-10 min vs. CFS/TGI-10 min, $P = 0.0413^*$, CFS/Sham vs. CFS/TGI-10 min, $P = 0.0460^*$). **e** The swimming length during training session. (RM two-way ANOVA, interaction: $P = 0.447$, time: $P < 0.0001^{***}$, treatment: $P = 0.0254^*$). **f** The swimming length on day 6 (two-way ANOVA, interaction: $P = 0.4233$, Control vs. CFS: $P = 0.0072^{**}$, Sham vs. TGI-10 min: $P = 0.0558$; post hoc Dunnett's test, Control/TGI-10 min vs. CFS/TGI-10 min, $P = 0.0356^*$, CFS/Sham vs. CFS/TGI-10 min, $P = 0.0465^*$). **g** Representative swimming paths for 2 min on day 7. **h** The percentage of time spent in the target quadrant (two-way ANOVA, interaction: $P = 0.1712$, Control vs. CFS: $P = 0.008^{**}$, Sham vs. TGI-10 min: $P = 0.2552$; post hoc Dunnett's test, Control/TGI-10 min vs. CFS/TGI-10 min, $P = 0.0167^*$, CFS/Sham vs. CFS/TGI-10 min, $P = 0.0074^{**}$). **i** A probe trial showing the total distance in the MWM (two-way ANOVA, interaction: $P = 0.1885$, Control vs. CFS: $P = 0.7202$, Sham vs. TGI-10 min: $P = 0.4574$). **j** Images depicting hippocampal neurons with anti-NeuN and a higher magnification CA1 area (Scale bars represent 250 and 50 µm). **k** Quantitative assessment of CA1 neuronal survival, Control/Sham ($n = 11$), Control/TGI-10 min ($n = 12$), CFS/Sham ($n = 12$), CFS/TGI-10 min ($n = 14$). (Two-way ANOVA, interaction: $P < 0.0001^{***}$, Control vs. CFS: $P < 0.0001^{***}$, Sham vs. TGI-10 min: $P < 0.0001^{***}$; post hoc Dunnett's test, Control/TGI-10 min vs. CFS/TGI-10 min, $P < 0.0001^{***}$, CFS/Sham vs. CFS/TGI-10 min, $P < 0.0001^{***}$). All data are displayed as means ± s.e.m

results suggested that the neuronal survival had no significant difference in LHA (Supplementary Fig. 1b), ML (Supplementary Fig. 1c) and SNc (Supplementary Fig. 1d) among Control, Susceptible and Susceptible/TGI-10 min groups.

Similarly, the expression of LC-TH$^+$ axons in the hippocampal CA1 region was quantified in CFS mouse model (Supplementary Fig. 2a–c). The results showed the pixel density of Th:LC-CA1 projections was also decreased in CFS mice, comparing with control mice (Supplementary Fig. 2d–g). Together, these findings indicated that the Th:LC-CA1 pathway is selectively decreased in CSDS-susceptible or CFS mice.

**Th:LC-CA1 circuit suppression mimics susceptibility effect.** Since CSDS accelerated TGI-induced hippocampal neuron and memory loss, we further tested whether suppressing the Th:LC-CA1 pathway alone may mimic depression-accelerated hippocampal neuron and memory loss upon TGI-10 min or not. Above all, the electrophysiological recording data demonstrated that activities of LC neurons in brain slices were either suppressed (hM4Di group) or enhanced (hM3Dq group) after CNO stimuli by using a multi-electrode array, while the spiking of LC neurons in mCherry-control group showed no significant response to CNO (Supplementary Fig. 3a–h). Then, we used the inhibitory DREADDs (Gi-coupled receptor, hM4Di) and its ligand Clozapine-N-oxide (CNO) to selectively silence the Th:LC-CA1 circuit. Bilateral co-injection of the AAV-TH-Cre and AAV-DIO-hM4Di-mCherry (or AAV-DIO-mCherry) viruses (Fig. 5a) into the LC enables the overexpression of the Gi-coupled inhibitory hM4Di receptor in TH$^+$-neurons, while the bilateral CA1-injection of CNO guarantees the selective silence of the Th:LC-CA1 circuit[44,45] (Fig. 5b). This experimental pipeline was illustrated in Fig. 5c. At the 28th day after viral injection, all mice received a bilateral CA1 CNO-injection in order to specifically inhibit Th:LC-CA1 projections before the TGI-10 min attack or sham-operated treatment (Fig. 5c). In the subsequent MWM test, hM4Di/TGI-10 min mice exhibited a significant worsening learning ability during the 6 days of the learning session and spent much more time to find the hidden platform at the 6th day (Fig. 5d–g) compared with mCherry/TGI-10 min mice. In the probe trial, hM4Di/TGI-10 min mice spent significantly less time in the target quadrant than mCherry/TGI-10 min mice (Fig. 5h, i), while the total swimming length was not different among all groups (Fig. 5j). Finally, we examined the effect of Th:LC-CA1 inhibition on CA1 neuron survival. Representative micrographs of NeuN-staining and statistical analysis demonstrated that numbers of CA1 pyramidal neurons in hM4Di/TGI-10 min mice

were significantly decreased compared with mCherry/TGI-10 min mice (Fig. 5k, l).

Additional experiment was performed to exclude the influence of CNO, which could be metabolised and affect the dopamine system, on the spatial memory and CA1 neuronal survival (Supplementary Fig. 4a–c). After 4 weeks of selective hM4Di-overexpression in LC-TH$^+$-neurons (Supplementary Fig. 4d–f), CNO or saline was stereotaxically injected in the CA1, then all mice were subjected to a TGI-10 min attack and MWM test. The results showed that the hM4Di/CNO mice displayed the worse learning ability (Supplementary Fig. 4g–j), memory performance (Supplementary Fig. 4k–m), and decreased CA1 neurons (Supplementary Fig. 4n, o), comparing to the Saline controls.

Taken together, these data demonstrated that suppression of Th:LC-CA1 circuit could mimic the role of chronic depression and worsen the TGI-induced spatial memory impairment and CA1 neuron loss.

**LC-CA1 circuit activation rescues CSDS-induced TGI injury.** To further confirm that CSDS-reduced Th:LC-CA1 projections are required for CSDS-induced hippocampal vulnerability to TGI-10 min, we specifically activated the Th:LC-CA1 circuit by utilising the excitatory DREADDs. Bilaterally injecting the AAV-Ef1a-DIO-hM3Dq-mCherry (or AAV-Ef1a-DIO-mCherry) and AAV-TH-Cre virus mixture (1:1) into the LC provides a Gq-coupled excitatory hM3Dq overexpression in the Th:LC neurons, while the bilateral injection of CNO into the CA1 guarantees the activation of the Th:LC-CA1 projection specifically[44,45] (Fig. 6a, b). The experimental pipeline of this process is further illustrated in Fig. 6c. After virus-injection, all mice were treated with a CSDS paradigm and then selected the susceptible phenotype individuals to subject sham-operated or TGI-10 min attack with CNO-injection. The four groups including mCherry/Sham, mCherry/TGI-10 min, hM3Dq/Sham and hM3Dq/TGI-10 min were subjected to MWM test and hippocampal CA1 neurons survival. Statistical analysis demonstrated that hM3Dq/TGI-10 min mice spent less time in the learning phase and swam short distances to reach a hidden platform compared with mCherry/TGI-10 min mice, and was no different compared with mCherry/Sham group (Fig. 6d–g). During the probe trial, hM3Dq/TGI-10 min mice spent significantly more time in the target quadrant to search for the hidden platform compared with the mCherry/TGI-10 min mice (Fig. 6h, i). The total swimming length among all groups was not significantly different (Fig. 6j), verifying that the increased platform-searching time in hM3Dq/TGI-10 min mice was a result of memory improvement. Consistent with the improved spatial learning and

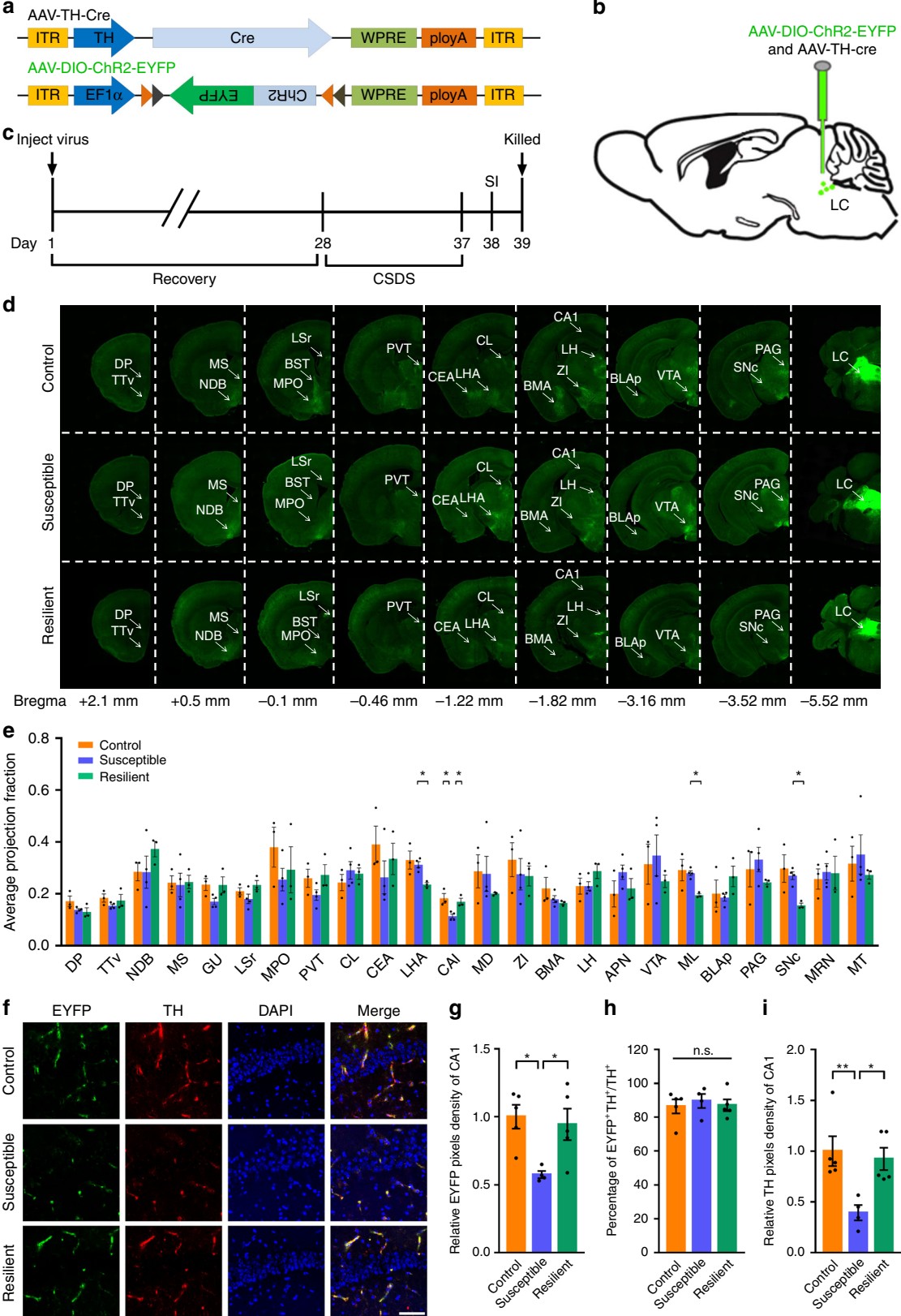

memory in hM3Dq/TGI-10 min mice, NeuN staining revealed that the numbers of surviving CA1 neurons were prominently and significantly increased in hM3Dq/TGI-10 min mice compared with mCherry/TGI-10 min mice (Fig. 6k, l). These results

demonstrated that activation of the Th:LC–CA1 pathway effectively rescued the TGI-induced spatial learning and memory impairment and CA1 neuronal loss as observed in CSDS-susceptible mice.

**Fig. 4** Whole-brain quantitative mapping of direct outputs from LC-TH+ neurons in CSDS. **a** Schematic maps of AAV vector constructs. **b** Schematic of Ef1α-DIO-hChR2(H134R)-EYFP combined with TH-Cre injection into the LC. **c** Experimental procedures. **d** Example of projections at nine coronal levels from LC-TH+ neuron populations. Measurements are given in millimetres from the bregma. Scaling bar = 1 mm. **e** Quantitative pixel density of output axons of LC-TH+ neurons in the corresponding brain area. Control ($n = 3$), Susceptible ($n = 4$), Resilient ($n = 3$). One-way ANOVA with post hoc Dunnett's test was used. DP ($P = 0.1501$), TTv ($P = 0.3787$), NDB ($P = 0.4218$), MS ($P = 0.9735$), GU ($P = 0.0999$), LSr ($P = 0.1521$), MPO ($P = 0.4466$), PVT ($P = 0.2107$), CL ($P = 0.5693$), CEA ($P = 0.4194$), LHA ($P = 0.0387$, Control vs. Susceptible, $P = 0.7548$, Susceptible vs. Resilient, $P = 0.0486*$), CA1 ($P = 0.013$, Control vs. Susceptible, $P = 0.0188*$, Susceptible vs. Resilient, $P = 0.0307*$), MD ($P = 0.5691$), ZI ($P = 0.7302$), BMA ($P = 0.3187$), LH (one-way ANOVA, $P = 0.2466$), APN ($P = 0.3067$), VTA ($P = 0.6144$), ML ($P = 0.0598$, Control vs. Susceptible, $P = 0.9012$, Susceptible vs. Resilient, $P = 0.0497*$), BLAp ($P = 0.2787$), PAG, periaqueductal grey ($P = 0.4653$), SNc ($P = 0.0576$, Control vs. Susceptible, $P = 0.7394$, Susceptible vs. Resilient, $P = 0.0481*$), MRN ($P = 0.9015$), MT ($P = 0.6952$). **f** Representative immunofluorescent staining shows the EYFP (green), anti-TH (red), and DAPI (blue) of CA1 region. Scaling bar = 250 μm. **g** Quantitative EYFP pixel density of directly projected axons of LC-TH+ neurons in CA1 area (one-way ANOVA, $P = 0.0111*$); post hoc Dunnett's test, Control vs. Susceptible, $P = 0.0156*$, Susceptible vs. Resilient, $P = 0.0330*$. Control ($n = 5$), Susceptible ($n = 6$), Resilient ($n = 5$). **h** Quantitation of EYFP+TH+ double positive in the whole TH+ signalling of CA1 region (one-way ANOVA, $P = 0.8476$). **i** Histogram depicting the TH+ signal of CA1 region (one-way ANOVA, $P = 0.0188*$; post hoc Dunnett's test, Control vs. Susceptible, $P = 0.0094**$, Susceptible vs. Resilient, $P = 0.0203*$). All data are displayed as means ± s.e.m

To elucidate the protective effect due to the chemogenetic enhancement of Th:LC-CA1 projections was dependent on a susceptible condition, the experiments including Resilient/Saline, Resilient/CNO, Susceptible/Saline and Susceptible/CNO groups were conducted. Briefly, all the mice were injected AAV-TH-Cre and AAV-DIO-hM3Dq-mCherry virus (Supplementary Fig. 5a–c), and then divided into either CSDS-resilient or -susceptible groups based on their different performances in the SI ratio (Supplementary Fig. 5d) and SI time in the interaction zone (Supplementary Fig. 5e). After receiving a Saline or CNO-injection in the CA1, all mice were subjected to a TGI-10 min attack and MWM test (Supplementary Fig. 5c). Consistently, the result of both MWM test and NeuN staining suggested that the Susceptible/Saline group merely displayed worse learning and memory performance (Supplementary Fig. 5f–l), and CA1 pyramidal neurons survival (Supplementary Fig. 5m, n) compared with other groups. However, activating the Th:LC-CA1 pathway in CSDS-resilient mice did not affect spatial learning and memory or neuronal survival (Supplementary Fig. 5f–n), which supports a specific role of the Th:LC-CA1 circuit in a CSDS-sensitive context.

**LC-CA1 circuit enhancement reverses CFS-induced TGI damage.** In accordance with the experimental procedures established in CSDS mice model (Supplementary Fig. 5), chemogenetic enhancement of Th:LC-CA1 projections was also performed in CFS-induced hippocampal vulnerability to TGI-10 min (Fig. 7a–c). After viral expression of Gq-coupled excitatory hM3Dq in the Th:LC neurons for 2 weeks, mice were subjected to a consecutive 14-day foot-shock protocol, which lead to a significant elevated in immobile time of FST (Fig. 7d). Saline or CNO-injection was administrated in the CA1 region before the TGI-10 min attack, then MWM test and CA1 neurons counting were followed. The results showed that, within the training session, all other groups except CFS/Saline group were no significance of the escape latency in the learning phase and the latency/swimming length to reach a hidden platform (Fig. 7e–h). Moreover, the probe trail within MWM test displayed that the CFS/CNO group spent significantly more time in the target quadrant to search for the hidden platform compared with the CFS/Saline mice (Fig. 7i–k). Consist with the improved spatial learning and memory in CFS/CNO mice, NeuN staining revealed that the numbers of surviving CA1 neurons were robustly and significantly elevated (Fig. 7l, m). The above results also confirmed that chemogenetic activation of the Th:LC-CA1 pathway rescues CFS-induced hippocampal vulnerability to TGI.

## Discussion

This study aimed to investigate the effect of depression on ischaemia-caused neurological disorders and to ascertain the underlying neural circuit mechanism. Our findings indicated that direct TH+-neuron projections from the LC to the CA1 served to mediate chronic stress (i.e., CSDS and CFS)-aggravated TGI-induced spatial memory impairment and CA1 neuronal death. Although numerous epidemiological studies have verified the association between depression and TGI, the detailed mechanisms of the neural circuit underlying this connexion remain largely unknown. Using virus-mediated whole-brain circuit mapping, we characterised the global distributions of LC neurons with cell-type specificity. Our result revealed that LC-TH+ neurons send axons to the CA1 region and CSDS treatment lessens this Th:LC-CA1 projection. Selective inhibition of the Th:LC-CA1 circuit using the DREADDs system mimicked the TGI-aggravated behavioural deficits and neuronal death in the absence of chronic stress. In addition, specific activation of this pathway significantly protected CSDS-susceptible and CFS-treated mice from ischaemic damage. These results provide a critical framework for understanding the downstream influence of the LC system in the CA1 region as it relates to spatial learning and memory.

Although epidemiological studies in several countries using nationwide cohorts have shown that depression may increases the risk of experiencing stroke[11,46–48], our findings further validate the relationship between depression and ischaemia in rodents. In this study, we employed chronic CSDS and CFS mouse models, but not an acute social defeat stress (single social defeat stress) or acute footshock stress model, as chronic and repeated stress protocols more accurately reflect the chronic symptoms and mechanisms of affective disorders such as depression. For example, the neuronal activity of the LC system has been shown to increase after acute social defeat stress, however, the spontaneous activity of LC neurons is lessened after prolonged stress treatment. In recent decades, the LC system has been the most extensively studied locus in response to stress. How the LC system, a classical NE neuron centre, serves to modulate neuronal activity in the hippocampus and affects spatial learning and memory has been the focus of several investigations[49,50]. Recent studies have discovered that the CA1 region in rodents is more preferentially innervated by neural fibres from the LC, providing a structural basis for these findings[26,27]. In this study, we validated the modulatory role of the LC system in the CA1 region, which effectively altered the tolerance of the CA1 region to ischaemia. Consistent with our hypothesis, inhibiting the LC-CA1 circuit appeared to reduce the release of neurotransmitters, thus increasing the vulnerability to ischaemic insult. The response of

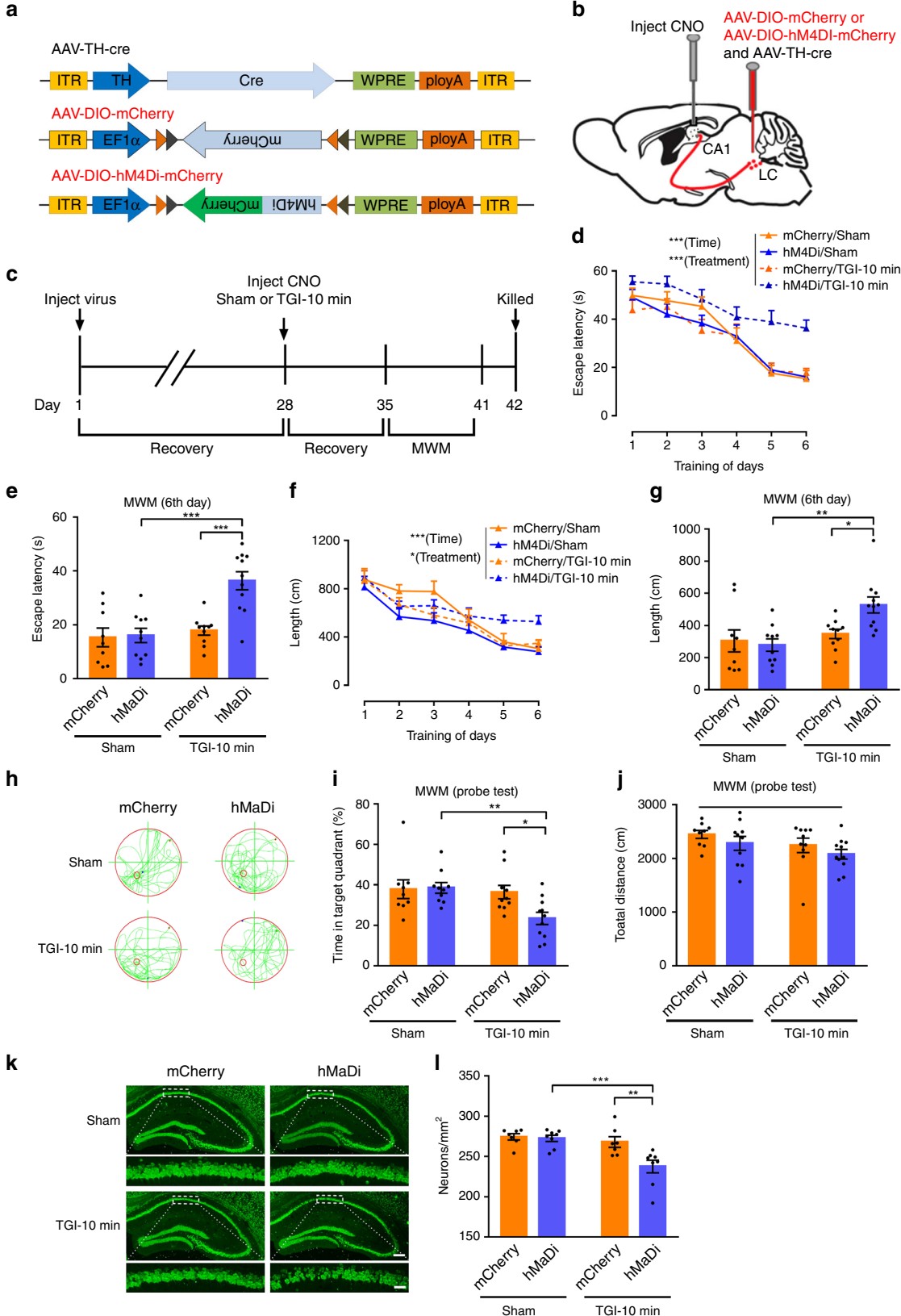

the LC-CA1 system to depression is particularly important in the context of stress-induced human neuropsychiatric disorders, including the cognitive impairment frequently observed in depression[51].

Dopamine neurotransmission in the dorsal hippocampus is crucial for spatial learning and memory[52,53]. Earlier studies have suggested that the TH[+] neuron population of the VTA is the presumed source of dopamine in the dorsal hippocampus[53–55].

**Fig. 5** Chemogenetic inhibition of Th:LC-CA1 circuit mimics susceptibility effect. **a** Schematic representation. **b** AAV-DIO-hM4Di-mCherry (or AAV-DIO-mCherry) and AAV-TH-Cre viruses were bilaterally injected into LC region. **c** Timeline of the experiments performed. **d** The escape latency of the MWM test (RM two-way ANOVA, interaction: $P = 0.1805$, time: $P < 0.0001$***, treatment: $P = 0.0002$***). mCherry/Sham ($n = 7$), mCherry/TGI-10 min ($n = 7$), hM4Di/Sham ($n = 8$), hM4Di/TGI-10 min ($n = 8$). **e** The latency to reach a hidden platform on day 6 (two-way ANOVA, interaction: $P = 0.0043$**, Sham vs. TGI-10 min: $P = 0.0004$***, mCherry vs. hM4Di: $P = 0.0022$**; post hoc Dunnett's test, mCherry/TGI-10 min vs. hM4Di/TGI-10 min, $P < 0.0001$***, hM4Di/Sham vs. hM4Di/TGI-10 min, $P < 0.0001$***). **f** The swimming length during the spatial learning trial (RM two-way ANOVA, interaction: $P = 0.2425$, time: $P < 0.0001$***, treatment: $P = 0.021$*). **g** Distribution of the swimming length on day 6 (two-way ANOVA, interaction: $P = 0.0376$*, Sham vs. TGI-10 min: $P = 0.004$**, mCherry vs. hM4Di: $P = 0.1153$; post hoc Dunnett's test, mCherry/TGI-10 min vs. hM4Di/TGI-10 min, $P = 0.0256$*, hM4Di/Sham vs. hM4Di/TGI-10 min, $P = 0.0016$**). **h** Representative swimming traces on day 7. **i** The percentage of time spent in the target quadrant (two-way ANOVA, interaction: $P = 0.0505$, Sham vs. TGI-10 min: $P = 0.0194$*, mCherry vs. hM4Di: $P = 0.0775$; post hoc Dunnett's test, mCherry/TGI-10 min vs. hM4Di/TGI-10 min, $P = 0.0227$*, hM4Di/Sham vs. hM4Di/TGI-10 min, $P = 0.0071$**). **j** Total distance of each group in the probe test (two-way ANOVA, interaction: $P = 0.9923$, Sham vs. TGI-10 min: $P = 0.0767$, mCherry vs. hM4Di: $P = 0.1486$). **k** Representative images of the hippocampal CA1 sub-region. (Green, NeuN$^+$, scale bars, 250 and 50 μm, respectively). **l** Quantification of CA1 pyramidal neurons. The statistical analysis was performed by two-way ANOVA (interaction: $P = 0.023$*, Sham vs. TGI-10 min: $P = 0.0016$**, mCherry vs. hM4Di: $P = 0.0108$*) with post hoc Dunnett's test (mCherry/TGI-10 min vs. hM4Di/TGI-10 min, $P = 0.0032$**, hM4Di/Sham vs. hM4Di/TGI-10 min, $P = 0.0006$***). mCherry/Sham ($n = 7$), mCherry/TGI-10 min ($n = 7$), hM4Di/Sham ($n = 8$) and hM4Di/TGI-10 min ($n = 8$). Data are displayed as means ± s.e.m

Our cell type-specific anterograde tracing technique revealed dense projections from LC-TH$^+$ neurons to the dorsal hippocampus in the physiological state. These results are consistent with the recent conclusion that LC-TH$^+$ neurons co-release dopamine to the dorsal hippocampus in order to promote spatial learning and memory[27]. During the past several decades, the LC system has been the most extensively studied in response to stress. In addition, several recent studies utilising innovative technologies have highlighted that evaluation of the whole-brain projection of LC neurons can contribute to a better understanding of its role in modulating diverse behaviours[53]. However, the alteration of the LC-TH$^+$ neural circuit in the progression of depression remains unclear. In this study, we detected that, with respect to CSDS treatment, TH-positive fibres from the LC to the CA1 region were significantly decreased in susceptible mice by employing whole-brain quantitative mapping. In conclusion, our findings in this study have illustrated the projection-selective mechanism of the complex LC system regarding diverse behavioural spectrums.

Despite the significance of the LC-CA1 circuit in mediating the vulnerability of CA1 neurons to ischaemic attack, the detailed mechanism of how LC-CA1 projecting neurons regulate CA1 local neural activity remains to be established. We hypothesise that certain efferent LC projections feed into particular synaptic receptors, which likely regulates the observed behavioural outcomes. For example, dopamine release from the LC to the CA1 has been shown to improve spatial object recognition and the enhancement of memory persistence via dopamine D1/D5 receptors[26,27]. Moreover, stress-accelerated defensive responses to looming are adrenergic receptor dependent[28]. Thus, further study is needed to elucidate what neurotransmitter (noradrenergic or dopamine) and associated receptors are the local effectors in the CA1 region. Recent studies have used optogenetic and chemogenetic technologies to manipulate brain circuits related to ischaemia[56,57]. In this study, we used chemogenetic technologies to regulate the circuit to determine how depression confers TGI-induced CA1 neuronal death and spatial memory dysfunction. The reason for this as it relates to chronic stress models is likely the character of stimulation, in which the light of the optogenetic method is transient, but the CNO drug of the chemogenetic method is enduring. Moreover, further study of how the electrophysiology and molecular profiles change in CA1 pyramidal neurons in response to depression is also warranted.

Our results in this study demonstrated that the TH$^+$ projection of LC neurons to the CA1 region is a functional efferent target for depression-induced aggravated responses to TGI. This dysfunction of the LC system is involved not only in psychiatric disorders, such as anxiety and depression, but it is also closely associated with neurodegenerative disorders, such as Alzheimer's disease and Parkinson's disease, with early and selective neural degeneration[58,59]. Therefore, we can take advantage of recent development tools, such as viral tracing and optogenetic and chemogenetic technologies to further determine the functional significance of the LC system, as well as the molecular mechanisms underlying the regulation of multiple behavioural consequences. Taken together, this study provides a fundamental framework for understanding the rodent brain circuitry responsible for depression aggravated, ischaemia-induced CA1 pyramidal neuron death and spatial memory impairment and suggests that further study of these brain pathways should greatly advance our current understanding of related psychopathologies.

## Methods

**Animals.** Adult (8 weeks) male C57BL/6J mice were used in this study and they were given access to food and water ad libitum. The mice were group-housed under a 12 h light-dark cycle (lights on at 8:00 a.m.), at consistent humidity (50 ± 5%), and ambient temperature (22 ± 1 °C). We have complied with all relevant ethical regulations for the animal testing and research in this study. All procedures were approved by institutional guidelines and the Animal Care and Use Committee (Huazhong University of Science and Technology, Wuhan, China) of the university's animal core facility.

**Transient global ischaemia.** The day before surgery, mice were placed in the operating room to adapt to the environment. Mice were anaesthetised with an intraperitoneal injection of chloral hydrate (350 mg/kg) and xylazine (10 mg/kg). The body temperature was monitored and maintained at 37 ± 0.5 °C using a feedback heating pad with all animals spontaneously breathing during the whole procedure. TGI was induced by bilateral common carotid artery occlusion. Briefly, an incision (1 cm) in the middle of ventral neck was made. Then the bilateral common carotid arteries were carefully isolated from the adjacent vagus nerve and tied with using a thin silk thread to cause the occlusion. After a period of ischaemia, the silk threads were removed and the arteries were visually inspected for reperfusion. Suturing the incision and disinfecting, mice were transferred to a warm environment for recovery approximately 2 h. Next, mice were returned to their home cages and housed with free access to food and water.

**Viral vectors.** AAV9 viruses encoding Ef1a-DIO-hM3Dq-mCherry, Ef1a-DIO-hM4Di-mCherry, Ef1a-DIO-mCherry, TH-Cre, Ef1α-DIO-hChR2(H134R)-EYFP were purchased from BrainVTA (BrainVTA Co., Ltd., Wuhan, China). The viral titres were in the range of $3–8 \times 10^{12}$ genome copies/ml. Viral vectors were subdivided into aliquots stored at −80 °C until use. Abbreviations of viral elements: ITR, inverted terminal repeat; DIO, double-flexed inverted open reading frame; WPRE, woodchuck hepatitis virus post-transcriptional regulatory element; Ef1a, human elongation factor-1 alpha promoter.

**Stereotaxic surgery.** Mice were anaesthetised with an intraperitoneal injection of chloral hydrate (350 mg/kg) and xylazine (10 mg/kg) and positioned in a

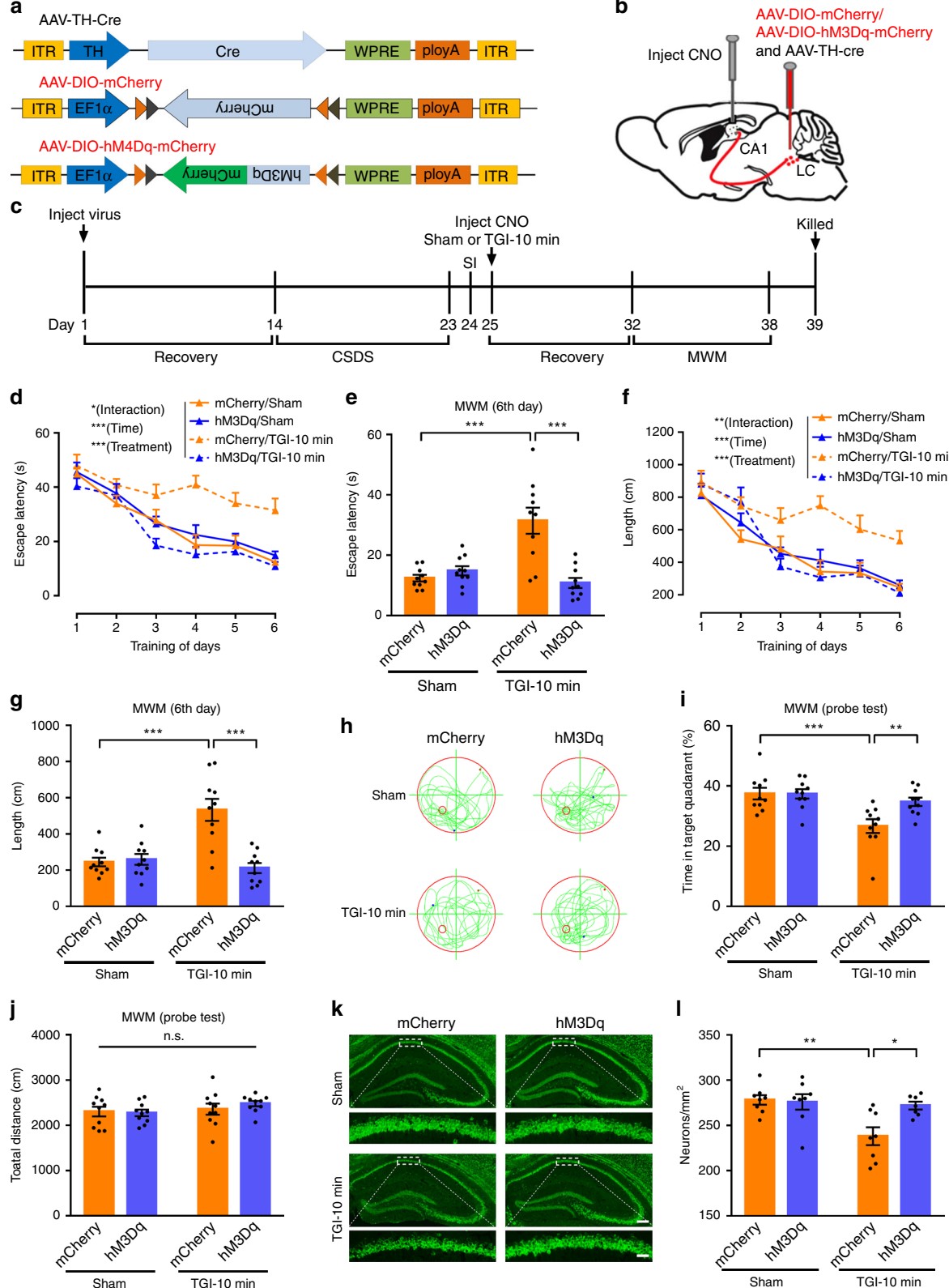

stereotaxic apparatus (68030, RWD, China). Body temperature was remained at 37 °C using a heating pad. Viruses were injected using a glass micropipette with a tip diameter of 15–20 μm, through a small skull opening (<0.5 mm²), with a quintessential stereotaxic injector from stoelting[44]. After each injection, the syringe was kept in place for 10 min to allow proper diffusion of the virus. Animals recovered on a heating pad until normal behaviour resumed. All

experiments involving viral constructs were performed after surgery to allow for sufficient expression. Viral infusion coordinates were LC (from bregma: anterior–posterior (AP), −5.3 mm, mediolateral (ML), 0.8 mm, and dorsal–ventral (DV) from the dura, −4.0 mm). For chemogenetic experiment, the LC received bilaterally injections. For whole-brain mapping experiment, the LC received unilateral injection.

**Fig. 6** Chemogenetic enhancement of Th:LC-CA1 circuit alleviates CSDS-worsened TGI. **a** Schematic representation of viral vectors. **b** AAV-DIO-hM3Dq-mCherry (or AAV-DIO-mCherry) and AAV-TH-Cre viruses were bilaterally injected into the LC region. **c** Experimental procedures. **d** Spatial learning curves in the MWM test (RM two-way ANOVA, interaction: $P = 0.0333*$, time: $P < 0.0001***$, treatment: $P < 0.0001***$). mCherry/Sham ($n = 10$), mCherry/TGI-10 min ($n = 10$), hM3Dq/Sham ($n = 10$), hM3Dq/TGI-10 min ($n = 10$). **e** The latency of mice to reach a hidden platform on day 6 (two-way ANOVA, interaction: $P < 0.0001***$, Sham vs. TGI-10 min: $P = 0.005**$, mCherry vs. hM3Dq: $P = 0.0009***$; post hoc Dunnett's test, mCherry/TGI-10 min vs. hM3Dq/TGI-10 min, $P < 0.0001***$, mCherry/Sham vs. mCherry/TGI-10 min, $P < 0.0001***$). **f** Summary of the distance during the 6-day training (RM two-way ANOVA, interaction: $P = 0.0055**$, time: $P < 0.0001***$, treatment: $P < 0.0001***$). **g** The swimming length on day 6 (two-way ANOVA, interaction: $P = 0.0001***$, Sham vs. TGI-10 min: $P = 0.0035**$, mCherry vs. hM3Dq: $P = 0.0003***$; post hoc Dunnett's test, mCherry/ TGI-10 min vs. hM3Dq/TGI-10 min, $P < 0.0001***$, mCherry/Sham vs. mCherry/TGI-10 min, $P < 0.0001***$). **h** Representative swimming traces on day 7. **i** Percentage of the time spent in the target quadrant on day 7 (two-way ANOVA, interaction: $P = 0.0326*$, Sham vs. TGI-10 min: $P = 0.0008***$, mCherry vs. hM3Dq: $P = 0.0359*$; post hoc Dunnett's test, mCherry/TGI-10 min vs. hM3Dq/TGI-10 min, $P = 0.0099**$, mCherry/Sham vs. mCherry/TGI-10 min, $P = 0.0006***$). **j** Total distance on day 7 (two-way ANOVA, interaction: $P = 0.4254$, Sham vs. TGI-10 min: $P = 0.1775$, mCherry vs. hM3Dq: $P = 0.6313$). **k** Representative fluorescent images of the CA1 area (green, NeuN$^+$) in mCherry/Sham, mCherry/TGI-10 min, hM3Dq/Sham and hM3Dq/TGI-10 min group. mCherry/ Sham ($n = 8$), mCherry/TGI-10 min ($n = 8$), hM3Dq/Sham ($n = 8$), hM3Dq/TGI-10 min ($n = 7$). **l** Quantification of CA1 pyramidal neurons. Statistical analysis was performed by two-way ANOVA (interaction: $P = 0.0239*$, Sham/TGI-10 min factor: $P = 0.0067**$, mCherry/hM3Dq factor: $P = 0.0446*$) with post hoc Dunnett's test (mCherry/TGI-10 min vs. hM3Dq/TGI-10 min, $P = 0.0114*$, mCherry/Sham vs. mCherry/TGI-10 min, $P = 0.0019**$). All data were displayed as means ± s.e.m

**In vivo chemogenetic manipulation**. For inhibition of chemogenetic experiments, the mixed virus containing Ef1a-DIO-hM4Di-mCherry (150 nl, $5.92 \times 10^{12}$ vg/ml) and TH-Cre (150 nl, $7.65 \times 10^{12}$ vg/ml) were injected into LC with coordinates of $-5.3$ mm AP, 0.8 mm ML and $-4$ mm DV. After 4 weeks of expression, WT mice were again anaesthetised and 1 μl Clozapine-N-oxide (CNO, Sigma, 1 μg/μl) or saline was bilaterally locally injected into the CA1 with coordinates of $-2.18$ mm AP, $\pm1.18$ mm ML ($+1.18$ mm ML on one side of CA1 and $-1.18$ mm ML on the other side of CA1 region) and $-1.36$ mm DV at a rate of 0.5 μl/min and subsequently subjected mice to TGI or sham-operated. Seven-day post-TGI operation, the behaviour test of MWM were carried out to test the ability of learning and memory. Then the mice were transcardially perfused with 0.9% saline followed by phosphate buffer (PBS) containing 4% paraformaldehyde. The expression of mCherry in the LC and CA1 pyramidal neuron were detected by immuno-fluorescence imaging with a microscope.

For experiments of chemogenetic activation, the mixed virus containing Ef1α-DIO-hM3Dq-mCherry (150 nl, $2 \times 10^{12}$ vg/ml) and TH-Cre (150 nl, $7.65 \times 10^{12}$ vg/ml) were injected into LC with coordinates of $-5.3$ mm AP, $\pm0.8$ mm ML and $-4.0$ mm DV at a rate of 30 nl/min. Two weeks later, mice underwent 10 days of CSDS or 14 days of CFS. Animals were again anaesthetised and 1 μl CNO or saline was bilaterally injected into the CA1 with coordinates of $-2.18$ AP, $\pm1.18$ ML and $-1.36$ DV at a rate of 0.5 μl/min. Mice were subjected to TGI operation without delay after CNO injection. MWM were tested after 7 days of TGI. Animals were deeply anaesthetised and perfused with 0.9% saline followed by 4% paraformaldehyde in 0.1 M PBS. The CA1 pyramidal neurons were detected by immunofluorescence imaging with a microscope.

**Whole-brain quantitative mapping**. For whole-brain mapping experiment of direct output with a 1:1 mixture of Ef1α-DIO-hChR2(H134R)-EYFP (100 nl, $7.65 \times 10^{12}$ vg/ml) and TH-Cre (100 nl, $7.65 \times 10^{12}$ vg/ml) unilaterally infused into LC region (AP $-5.3$ mm, ML $\pm0.8$ mm and DV $-4$ mm) at a rate of 20 nl/min. Four weeks later, mice underwent 10 days of CSDS. The day prior to sacrifice, animals underwent one round of SI test to ascertain the resilient or susceptible mice. The method of quantitation and statistic was according to the previous studies[42,60]. Briefly, all the mice were perfused with paraformaldehyde and treated with the gradient of sucrose for cryoprotection, then performed the 40 um thickness frozen slices at nine levels of bregma 2.1, 0.5, $-0.1$, $-0.46$, $-1.22$, $-1.82$, $-3.16$, $-3.52$ and $-5.52$ mm. The whole-brain sections were used by automatic scanning fluorescence microscope (Olympus, SV120) for imaging. Then, the pixel density was analysed by Image J project (Fiji version) from NIH image[61] for the corresponding brain area (including, dorsal peduncular area; ventral part of taenia tecta; diagonal band nucleus; medial septal nucleus; gustatory areas; rostral part of lateral septal nucleus; medial preoptic area; paraventricular nucleus of the thalamus; central lateral nucleus of the thalamus; central amygdalar nucleus; lateral hypothalamic area (LHA); mediodorsal nucleus of thalamus; zona incerta; basomedial amygdalar nucleus; lateral habenula; anterior pretectal nucleus; VTA; ML; posterior part of basolateral amygdalar nucleus; periaqueductal grey; SNc; midbrain reticular nucleus; medial terminal nucleus of the accessory optic tract), which was aligned by the mouse brain map of Allen Brain Atlas[62]. Meanwhile, the pixel density of viral injection site with LC area was also quantitated. The projection fraction was calculated as the ratio of corresponding brain area pixel density to LC region. Moreover, at least three contiguous sections were scanned to calculate the average projection fraction for each brain area in every mouse. Finally, the average projection fraction was statistically analysed in control, susceptible and resilient groups.

**Chronic social defeat stress**. CSDS protocol was carried out according to the model described by Golden et al.[41]. Briefly, prior to experiment CD1 mice were screened on three consecutive days and selected according to the following criteria: (i) a latency of attack under 1 min and (ii) attacking for 2 consecutive days. For 10 consecutive days, experimental male C57BL/6J mice (intruder) were subjected to physical interaction and defeat by the aggressive CD1 (resident) in resident home cage for 5 min (one defeat stress per day). After physical defeat, the animal remained in the aggressor's home-cage on the other side of a perforated translucent Plexiglas divider to allow visual, auditory and olfactory interaction between intruder and resident for 24 h until the next bout of physical defeat. Intruder mice always faced novel CD1 mice. The control mice (or undefeated mice) were housed together separated by the Plexiglas divider barrier and switched each day. They were never exposed to the CD1. After 10 days of CSDS, the experimental mice were singly housed and tested 24 h later for social avoidance behaviour.

**SI test**. This test was measured using the two-phase SI test. In the first phase, animals were placed in an open field (50 cm × 50 cm × 50 cm) with an empty Plexiglas wire mesh enclosure (10 cm wide × 6.5 cm deep × 42 cm high). The time spent in the interaction zone surrounding the wire mesh enclosure was measured over 2.5 min during the first phase in which the wire mesh with empty (no target). Animals were then returned to home cage for 30 s. In the second phase, an unfamiliar, aggressive CD1 mouse was placed inside the wire mesh (target) and the same metrics were measured. From these two stages, an SI index was calculated (SI is equal to the times spent in interaction zone with target divided by times spent in interaction zone with no target). Animals were defined as susceptible if SI value less than 1, and resilient if SI greater than 1.

**Chronic footshock stress**. Animals were assigned to experimental groups randomly. The control mice were left undisturbed and caged in groups of three. The animals in the CFS group were caged singly and subjected to 14 days or more of daily footshock session through a grid floor of the shock boxes according to the previous publication[63]. Briefly, a random shock generator was used to deliver foot shocks of 0.5 mA within a 20 min period. The duration of each shock and the intervals between the adjacent shocks was randomly programmed between 1 and 3 s and 1 and 15 s, respectively.

**Morris water maze**. A tank 1.2 m in diameter filled with water was made opaque with a white, nontoxic ink maintained at 23–25 °C. The maze was labelled with several distinct extra-maze cues. A camera, fixed to the device with 1 m from the water surface, was connected to a digital tracking device (Shanghai Xinruan Information Technology Co. Ltd). A computer with the Morris water maze software then processes the tracking information. The animals were handled for 2–3 days[64] and were kept on outer-room shelves to eliminate auditory cues and directional olfactory. They were brought to the testing room 2 h before test. The first day they were training by a visible platform (10 × 10 cm). If a mouse did not find the platform within 60 s, it was guided to the platform by the experimenter and was allowed to remain on the platform for 45 s. Next mice underwent their first hidden platform training with the platform submerged 1 cm below the surface. The platform location remained the unchanged throughout the training, but the start location varied randomly during the four daily trials (north, south, east or west). Mice received four trails per day for 6 consecutive days with a 15 min inter-trial interval. The maximum time allowed per trial was 60 s. If a mouse did not find or mount the platform, it was guided to the platform by the experimenter. All mice were allowed to remain on the platform for 45 s before being removed by the

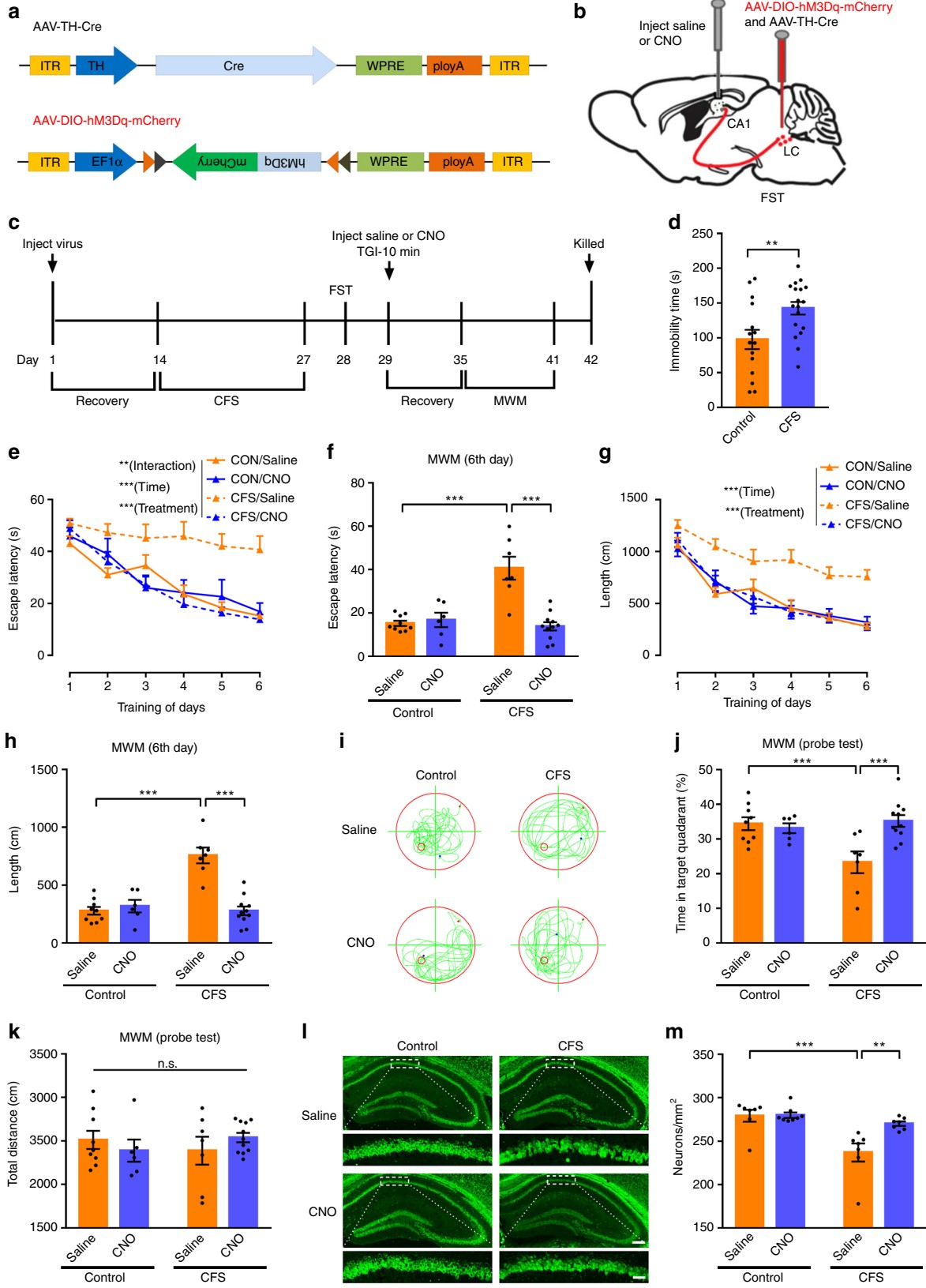

experimenter. The total training took 6 days and then followed by the probe test on the seventh day. For probe trials, the platform was removed, and each mouse was allowed to swim for 120 s. The escape latency, distance travelled, time in target quadrant and the total distance were recorded automatically for following analysis.

**Forced swimming test**. Mice were placed in 20 cm of water (24 ± 2 °C) in Plexiglas cylinders (30 cm height × 15 cm diameter) for 6 min The behaviour of mice was recorded by a video-tracking system (Shanghai Xinruan Information Technology Co. Ltd., China) and the immobility time was recorded automatically during the session of 6 min. Mice were considered immobile when it remained floating in the

**Fig. 7** Th:LC-CA1 circuit activation relieves CFS-induced TGI susceptibility. **a** Schematic representation of the construct of the AAV-DIO-hM3Dq-mCherry and AAV-TH-Cre virus. **b** The mixture viruses including AAV-DIO-hM3Dq-mCherry and AAV-TH-Cre were bilaterally injected into LC area. **c** Experimental timeline. **d** The immobility time by FST. Control ($n = 15$), CFS ($n = 18$). Unpaired two-tailed Student's $t$ test, $P = 0.0086$**. **e** Spatial learning curves in the MWM test (RM two-way ANOVA, interaction: $P = 0.0029$**, time: $P < 0.0001$***, treatment: $P < 0.0001$***). **f** Escape latency on day 6. Control/Saline ($n = 9$), Control/CNO ($n = 6$), CFS/Saline ($n = 7$), CFS/CNO ($n = 11$). Two-way ANOVA, interaction: $P < 0.0001$***, Control vs. CFS: $P = 0.0007$***, Saline vs. CNO: $P = 0.0002$***; post hoc Dunnett's test, CFS/Saline vs. CFS/CNO, $P < 0.0001$***, Control/Saline vs. CFS/Saline, $P < 0.0001$***. **g** Total distance during the 6-day training (RM two-way ANOVA, interaction: $P = 0.2195$, time: $P < 0.0001$***, treatment: $P < 0.0001$***). **h** Swimming length on day 6. Two-way ANOVA, interaction: $P < 0.0001$***, Control vs. CFS: $P < 0.0001$***, Saline vs. CNO: $P < 0.0001$***; post hoc Dunnett's test, CFS/Saline vs. CFS/CNO, $P < 0.0001$***, Control/Saline vs. CFS/Saline, $P < 0.0001$***. **i** Representative swimming traces on day 7. **j** Percentage of the time spent in the target quadrant on day 7. Two-way ANOVA, interaction: $P = 0.0045$**, Control/CFS factor: $P = 0.0444$*, Saline/CNO factor: $P = 0.0195$*; post hoc Dunnett's test, CFS/Saline vs. CFS/CNO, $P = 0.0009$***, Control/Saline vs. CFS/Saline, $P = 0.0027$**. **k** Total distance on day 7. Two-way ANOVA, interaction: $P = 0.2149$, Control vs. CFS: $P = 0.9044$, Saline vs. CNO: $P = 0.9001$. **l** Representative fluorescent images of the CA1 region (NeuN$^+$ as green). **m** Quantification of CA1 neuronal survival. Control/Saline ($n = 7$), Control/CNO ($n = 8$), CFS/Saline ($n = 7$), CFS/CNO ($n = 7$). Two-way ANOVA, interaction: $P = 0.0182$*, Control vs. CFS: $P = 0.0004$***, Saline vs. CNO: $P = 0.0143$*; with post hoc Dunnett's test, CFS/Saline vs. CFS/CNO, $P = 0.0039$**, Control/Saline vs. CFS/Saline, $P = 0.0003$***. Data were displayed as means ± s.e.m

water, without struggling, making only very slight movements necessary to keep its head above water[65].

**Open field test**. An open field area (50 cm × 50 cm × 50 cm) made of white PVC was used to assess spontaneous activity. After acclimation to the testing room for 2 h, mice were tested for 10 min in the open field area. A video-tracking system (Shanghai Xinruan Information Technology Co. Ltd.) was used to measure the spontaneous activity of the animal. Total distance travelled in open filed area (total activity) were analysed.

**Rotarod test**. A rotating cylinder (6 cm in diameter) with a coarse surface for a firm grip was used to assess motor coordination. After acclimation to the testing room for 2 h, mice were placed onto the central with the speed of the rotarod accelerated from 5 to 10 rpm and given a 30 min training session. After the training, the speed of the rotarod accelerated from 5 to 40 rpm over a 5 min period. The time at which each mouse falls from the rotating rod was automatically recorded by the detector. The device was cleaned between mice with 75% ethanol solution. Mice underwent four trails a day with 30 min rest each trail.

**Grip strength test**. The grip strength test is designed to evaluate muscle strength in vivo, performed by gently lifting the mouse's tail to assess its tendency to grasp a grid. The grip test is used to assess the maximal muscle strength of the four limbs. Mice were placed on a 23 × 25 cm grid and were gently pulled horizontally backward until they released their grip[66]. A grip strength metre (Shandong Academy of Medical Sciences, China) attached to a force sensor used to measure the peak force formed by the four limbs. Untrained mice were tested four times without rest and the maximal strength developed by each animal (in grams) was recorded.

**Four-limb hanging test**. The four-limb hanging test (also known as Kondziella's inverted screen test) is used to assess muscle strength using all four limbs and endurance over time in mice. Place the mice in the central of the wire grid screen and rotate the screen to an inverted position with the mouse's head declining first. Hold the screen steadily 40–50 cm above a padded surface[67]. This assay makes use of a wire grid system to noninvasively measure the ability of mice to exhibit sustained limb tension to oppose their weight. The time spent hanging was measured until they fall.

**Immunofluorescence**. Coronal sections (20–40 μm) of mouse brains fixed in 4% (wt/vol) paraformaldehyde were prepared via a cryostat microtome. Antigen retrieval was performed by using 1% sodium dodecyl sulphate (SDS) in PBS. Following rinsing with PBS, the non-specific protein binding was blocked for 1 h with 10% goat-serum and 0.5% Triton-X in PBS. Mouse monoclonal anti-NeuN antibody (1:100, MAB377, Millipore) and rabbit polyclonal anti-TH (1:200, 25859-1-AP, Proteintech) were applied diluted in 5% bovine serum albumin (BSA) in PBS and incubated overnight at 4 °C. After rinsing three times in PBS for 5 min each, the sections were incubated in fluorochrome-conjugated secondary antibody (1:400, Dylight-488-labelled goat anti-mouse or goat anti-rabbit, Abbikine, CA, USA) diluted in PBS for 1 h at room temperature in the dark. Then rinsing three times in PBS for 5 min DAPI (1:1000, D9542, Sigma-Aldrich) used to validate the morphological identification of nucleus. The sections were cover-slipped with fluorescent mounting medium and stored in 4 °C before analysis. Immunofluorescence evaluation was performed by fluorescence imaging using the Olympus IX-73 microscope connected to Olympus DP80 photographic equipment (Olympus, Japan). For confocal microscopy, images were collected on a Carl Zeiss LSM780 laser scanning confocal microscope (Zeiss Microsystems, Jena, Germany).

**Statistical analysis**. All the results were displayed as the mean ± s.e.m. with *$P < 0.05$, **$P < 0.01$, ***$P < 0.001$, and n.s. represents no significant. We performed the unpaired two-tailed Student's t-test for the experiments with only 2 groups, one-way analysis of variance (ANOVA) with post hoc Dunnett's test for the single factor experiments with > 2 groups, two-way ANOVA followed by multiple comparisons with post hoc Dunnett's test for the double factor experiments. In the water maze test, we performed the repeated-measures (RM) ANOVA for the training phase (6 days), and ordinary two-way ANOVA followed by multiple comparisons with post hoc Dunnett's test for the sixth-day escape latency, length and probe test. All statistical analyses were performed using the Graphpad Prism software.

## Data availability
The data that support the findings of this study are available from the corresponding author upon reasonable request.

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

## Acknowledgements

This work was supported financially by grants from National Natural Science Foundation of China (Nos. 31371384 and 31571044 to B.T., Nos. 81471386 and 81672504 to X.Q.C. and No. 31600821 to P.Z.), Programme for New Century Excellent Talents in University (No. NCET-10-0415 to B.T.), Integrated Innovative Team for Major Human Diseases Programme of Tongji Medical College, HUST (Grant 5001530026), China Postdoctoral Scientific Foundation (No. 2015M582226 and 2018T110774 to P.Z.), Natural Science Foundation of Hubei Province (No. 2017CFB465 to P.Z.), Educational Commission of Hubei Province of China (No. D20182102 to P.Z.) and the Fundamental Research Funds for the Central Universities (HUST: 2019kfyXJJS081 to P.Z.).

## Author contributions

Q.Z., D.X.H., F.H. and C.Y.L. performed the animal studies. Q.Z., C.Y L., D.X.H., F.H., G.Q. and H.W.C. performed the behaviour test. Q.Z., D.X.H., F.H., C.Y.L., T.X.L. and P.Z. performed the fluorescence experiments. Q.Z., D.X.H., J.M., P.Z., X.Q.C. and B.T. designed the research, analysed the data, wrote and edited the paper.

## Additional information

**Competing interests:** The authors declare no competing interests.

