## [Transparent Peer Review File · Nature Communications]

Reviewers' comments:

Reviewer #1 (Remarks to the Author):

In this study, the authors demonstrate a role for Th+ locus coeruleus-CA1 projections in depression-induced vulnerability to hippocampal neuronal loss and cognitive impairments following transient global ischemia (TGI). The authors characterize a model of TGI at increasing lengths of ischemia, demonstrating that TGI induced for 20-30 minutes leads to selective CA1 neuronal death and spatial memory impairment as assessed via Morris water maze (MWM), while a 10 minute TGI shows no effects on hippocampus impairment. Depression was modeled in mice using chronic social defeat stress (CSDS) and chronic footshock stress (CFS), with susceptible CSDS mice and CFS mice showing cognitive impairment and CA1 neuronal death following 10 min TGI, indicating an exacerbated response to the ischemic event. The Th:LC-CA1 pathway is identified as critical for mediation of depression-induced TGI morbidity via assessment of LC-TH+ neuron projections in CSDS susceptible mice. The authors demonstrate that chemogenic inhibition of the Th:LC-CA1 pathway induces loss of CA1 neurons and spatial memory impairment following 10 min TGI (similar to susceptible CSDS and CFS mice), while chemogenic activation of the pathway rescues aggravated TGI responses after CSDS. Taken together, these findings demonstrate a role for the Th:LC-CA1 pathway in depression-induced vulnerability to TGI.

Significance

Meta analyses of epidemiological studies have indicated that depression is associated with a significantly increased risk of stroke morbidity and mortality. (JAMA 2011) The findings of this study are a significant contribution to the field as it is the first to model increased stroke morbidity in depressed mice. However, it is important to note that this model, one of global ischemia, is not the "stroke" that most physicians and scientists have in mind when they use this term. Instead, for the neuroscience community and in terms of epidemiology, "stroke" means focal ischemic stroke. The manuscript repeatedly overstates its relevance to stroke (see below). Nonetheless, these findings provide strong evidence for a novel Th:LC-CA1 in the pathology of depression and a very much less common form of ischemic stroke.

Major Concerns

1. The authors identify the Th:LC-CA1 pathway as critical for mediation of depression-induced TGI morbidity via assessment of LC-TH+ neuron projections in CSDS susceptible mice. This finding would be strengthened through quantification of this pathway in CFS mice as well.

The findings presented in this study indicate a role for the Th:LC-CA1 pathway in depression-induced response to TGI. The experiments presented in Figure 5 demonstrate that DREADD inhibition of these projections mimics the spatial and memory impairment and CA1 neuronal death observed in depressed mice following TGI, while the results in Figure 6 indicate that DREADD activation of the circuit rescues the spatial memory impairment and CA1 neuronal death induced by depression and TGI. The conclusions of each of these experiments would be further strengthened by the inclusion of CNO/Sham groups. This would indicate whether or not the role of this pathway in cognition and CA1 neuronal survival is dependent on an ischemic condition.

2. The findings of the chemogenic activation experiments would be strengthened by demonstration of rescue in the CFS mice as well as the susceptible CSDS mice.

3. The statistics are not described in the Methods, and in crucial areas (such as in Figs 2 and 3) appear to be T tests between a control group and depression group in TGI. However, one cannot do a T test in a study with >2 groups, and the testing in this manuscript does not appear to be appropriate.

4. The quantification Method for the Th+ neuronal arbors from the LC is not described—just that

these arbors were imaged and then aligned with an Allen Brain atlas.

5. In the DREADD studies, it is now well recognized that CNO is metabolized to clozapine, which directly acts on the dopamine system—which is the system under study in this manuscript. The authors need to comment on the specificity of their work, and how it established that the gain and loss of function are due to DREADD GPCR activation, and not clozapine-dopamine receptor blockade.

Minor Concerns

Though the authors state at the end of the introduction that TGI-10min was used for all studies, the alternate labeling of TGI (Fig 2-3) vs TGI-10 (Fig 5-6) is potentially confusing to the reader.

Reviewer #2 (Remarks to the Author):

In this study, Zhang et al., show that depression in mice, caused by chronic social defeat stress (CSDS) or chronic footshock stress (CFS), induced an elevated response to TGI that caused memory impairment and was found to be mediated by the Th:LC-CA1 pathway. In addition, they show that selective chemogenetic inhibition of TH+ projections from the LC to the CA1 mimics depression-like behaviors and induces a worsening response to TGI. Furthermore, the authors claim that chemogenetic activation of the Th:LC-CA1 pathway can reverse these aggravated TGI responses after CSDS. This is a very interesting and laborious study that provides novel insights into the mechanisms of the interplay between depression and TGI and how this affects hippocampal-dependent memory. However, there are also several aspects of the work that are problematic and are presently unconvincing. Validation and control experiments are warranted to strengthen the interpretation of the overall hypothesis.

1. It is not clear how statistical analysis of the water maze training data (6 days) is performed. The authors mention a two-way ANOVA analysis. However, a repeated-measures ANOVA should be performed. In addition, statistical values need to be reported. Is RM-ANOVA significant and for what parameters? In case of significance, which post-hoc analysis is performed? By looking at the graphs, in most cases there are differences on day 6, but I am not sure that this is enough to justify significance in RM-ANOVA. The authors should report all statistical values better and do that analysis more carefully.

2. Following the chronic social defeat stress paradigm, do the authors observe comparable proportions of susceptible and resilient mice as described in previous reports (e.g. Golden et al, 2011, Nat Protoc)? If not, this should be explained. Additionally, the authors write, "As expected, susceptible mice showed marked decreases in social avoidance in the presence of CD1 mice", which is in conflict with the displayed data. The correct phrasing of "decreased social interaction" or "increased social avoidance" should be used.

3. The authors use a hChR2-YFP fusion protein to label axons originating from the LC in the CA1 region. However, the axonal labeling in their high magnification panels (Fig. 4f) does not look convincing. More needs to be done to prove this signal truly represents LC terminals. To be more convincing, they should co-stain these sections with an anti-TH antibody and show colocalization, and then quantify co-staining with TH to know only LC terminals are stained. Moreover, if their claim is true, a reduction in overall TH signal should be apparent in their susceptible brains (given that TH+ axons from the VTA should be much less abundant than those from the LC axons).

4. Their chemogenetic activation data are also problematic. In the water maze probe trial, the saline/susceptible group shows no significant memory deficits compared with the resilient groups. This result is a major issue and does not agree with the initial hypothesis. Although CNO treatment clearly shows memory improvement, it looks like it is also significantly different from the resilient

group. This difference is mainly due to the poor performance of all other groups (about 25% of time spent in target quadrant, something you see also in Fig. 2k). Again, the statistical details should be provided, and comparisons need to be made between all groups.

5. A control experiment is absent in Figures 5 and 6. CNO should be injected in CA1 of animals that received LC infection of mCherry-only expressing virus. This is necessary to demonstrate specificity of CNO administration on DREADD activation or inhibition of LC terminals. Moreover, functional expression of the DREADDs is not demonstrated using either electrophysiological recording of LC neurons with bath application of CNO or with microdialysis (or comparable sampling methodology) to assay suppression or enhancement of NE/DA release in CA1 with CNO application.

4. Why do the authors use Nissl staining in Figure 3, while they use NeuN immunoreactivity in all other instances? They have to show quantification of NeuN in Figure 3 as well.

5. Panels of Figure 2c-d are mislabeled in the text as 1c-d. Also in Figure 2d target/no target labels should be inverted

More Minor issues:

6. A non-trivial percentage of LC neurons which were TH-negative expressed mCherry after viral infection (~10%). Can the authors comment on this population? Were the projection targets of the mCherry+ TH- LC neurons identified? What is the neurotransmitter synthesized/released from these neurons?

7. Resilient mice appear to demonstrate diminished LC-derived TH staining in LHA, ML, and SNc. What does this mean? Is neuronal survival negatively impacted in these regions after the chronic stress and TGI as observed in the susceptible mice after CSDS?

8. The methodology is well outlined, but there are numerous grammatical errors. This section, in particular, requires further proof-reading and editing by the authors.

Point-by-point response to the referees' comments

Response to Reviewer comment # 1:

Reviewer #1 (Remarks to the Author):

In this study, the authors demonstrate a role for Th+ locus coeruleus-CA1 projections in depression-induced vulnerability to hippocampal neuronal loss and cognitive impairments following transient global ischemia (TGI). The authors characterize a model of TGI at increasing lengths of ischemia, demonstrating that TGI induced for 20-30 minutes leads to selective CA1 neuronal death and spatial memory impairment as assessed via Morris water maze (MWM), while a 10 minute TGI shows no effects on hippocampus impairment. Depression was modeled in mice using chronic social defeat stress (CSDS) and chronic footshock stress (CFS), with susceptible CSDS mice and CFS mice showing cognitive impairment and CA1 neuronal death following 10 min TGI, indicating an exacerbated response to the ischemic event. The Th:LC-CA1 pathway is identified as critical for mediation of depression-induced TGI morbidity via assessment of LC-TH+ neuron projections in CSDS susceptible mice. The authors demonstrate that chemogenic inhibition of the Th:LC-CA1 pathway induces loss of CA1 neurons and spatial memory impairment following 10 min TGI (similar to susceptible CSDS and CFS mice), while chemogenic activation of the pathway rescues aggravated TGI responses after CSDS. Taken together, these findings demonstrate a role for the Th:LC-CA1 pathway in depression-induced vulnerability to TGI.

Significance

Meta analyses of epidemiological studies have indicated that depression is associated with a significantly increased risk of stroke morbidity and mortality. (JAMA 2011) The findings of this study are a significant contribution to the field as it is the first to model increased stroke morbidity in depressed mice. However, it is important to note that this model, one of global ischemia, is not the “stroke” that most physicians and scientists have in mind when they use this term. Instead, for the neuroscience community and in terms of epidemiology, “stroke” means focal ischemic stroke. The manuscript repeatedly overstates its relevance to stroke (see below). Nonetheless, these findings provide strong evidence for a novel Th:LC-CA1 in the pathology of depression and a very much less common form of ischemic stroke.

Response: We greatly appreciate for the reviewer's positive comments on the significance of this study. We thank the reviewer's kind remind for the proper use of the term “stroke” in the manuscript. We have replaced the term “stoke” with “transient global ischaemia” when it refers to the ischemic model in the manuscript.

Major Concerns

1. The authors identify the Th:LC-CA1 pathway as critical for mediation of depression-induced TGI morbidity via assessment of LC-TH⁺ neuron projections in CSDS susceptible mice. This finding would be strengthened through quantification of this pathway in CFS mice as well.

The findings presented in this study indicate a role for the Th:LC-CA1 pathway in depression-induced response to TGI. The experiments presented in Figure 5 demonstrate that DREADD inhibition of these projections mimics the spatial and memory impairment and CA1 neuronal death observed in depressed mice following TGI, while the results in Figure 6 indicate that DREADD activation of the circuit rescues the spatial memory impairment and CA1 neuronal death induced by depression and TGI. The conclusions of each of these experiments would be further strengthened by the inclusion of CNO/Sham groups. This would indicate whether or not the role of this pathway in cognition and CA1 neuronal survival is dependent on an ischemic condition.

Response: We greatly appreciate the reviewer's valuable comments and agree that the quantification of the Th:LC-CA1 pathway in CFS and the inclusion of CNO/Sham groups in Figure 5 and Figure 6 are important to strengthen our findings. We have completed all these experiments and the results supported our proposed role of the Th: LC-CA1 pathway in depression-induced response to TGI.

In Supplementary Figure 2a-c, we have quantified the Th: LC-CA1 projection in chronic foot shock (CFS) mice by using cell-specific tracing technique of direct output projections, as it was done in CSDS-susceptible mice. The result suggested that CFS group mice showed a significant decrease of LC-TH⁺ axons in the CA1 region, comparing with the control mice (Supplementary Figure 2d, e). Meanwhile, the proportions of EYFP⁺TH⁺ in the whole TH⁺ population was conducted in all groups (Supplementary Figure 2f), and the reduction of overall TH⁺ signal (Supplementary Figure 2g) of CA1 region was also apparent in mice subjected to CFS treatment.

In revised Figure 5, we have re-conducted the experiments in order to include CNO/Sham groups. Briefly, the mice were divided into four groups including mCherry/Sham, mCherry/TGI-10min, hM4Di/Sham, and hM4Di/TGI-10min in chemogenetic suppression experiment. All mice were subjected to CNO with brain stereotactic injection. The experimental pipeline was the same as that in original Figure 5. The results demonstrated that DREADD inhibition of the Th:LC-CA1 pathway in hM4Di/TGI-10min mice mimics the spatial and memory impairment and CA1 neuronal death observed in depressed mice following TGI. Original Figure 5 has been moved to Supplementary Figure 4 in the current manuscript, which suggested that hM4Di-overexpression alone in LC region (saline group) was ineffective in the chemogenetic inhibition of the Th:LC-CA1

pathway.

In revised Figure 6, we have conducted additional DREADD activation-rescue experiments in order to include CNO/Sham groups. The mice were divided into four groups including mCherry/Sham, mCherry/TGI-10min, hM3Dq/Sham, and hM3Dq/TGI-10min. All mice were stereotactically injected with CNO after the CSDS treatment and the experimental pipeline was the same as that in original Figure 6. Meanwhile, original Figure 6 has been moved to Supplementary Figure 5 in the current manuscript, which suggested that spatial memory and CA1 pyramidal neurons survival were prominently and significantly rescued in Susceptible/CNO mice compared with Susceptible/Saline mice in the chemogenetic enhancement of the *Th*:LC-CA1 pathway.

These evidences indicate that the role of *Th*: LC-CA1 projection in cognition and CA1 neuronal survival is dependent on an ischemic condition.

2. The findings of the chemogenic activation experiments would be strengthened by demonstration of rescue in the CFS mice as well as the susceptible CSDS mice.

Response: Thanks for the reviewer's important suggestions. We have completed the chemogenetic activation-rescue experiments in CFS mice and demonstrated that the activation of *Th*: LC-CA1 projections could rescue cognition and CA1 neuronal survival in the CFS mice (Figure 7). The experimental pipeline of rescue in the CFS mice (Figure 7a-c) was similar to that in the susceptible CSDS mice (Supplementary Figure 5). Briefly, the mice were divided into four groups including CON/Saline, CFS/Saline, CON/CNO, and CFS/CNO groups. All mice were infected with hM3Dq for 2 weeks in the *Th*:LC neurons by bilateral stereotaxic injection of virus, then CFS group mice were subjected to a consecutive 14 day-foot-shock protocol, which lead to a significant elevated in immobile time of forced swimming test (FST) (Figure 7d). Saline or CNO-injection was administrated in the CA1 region before the TGI-10min attack or sham operation, then MWM test and counting of CA1 neurons were followed (Figure 7c). The results showed that, within the training session, all other groups except CFS/Saline group were no significance of the escape latency in the learning phase and the latency/swimming length to reach a hidden platform (Figure 7e-h). Moreover, the probe trail within MWM test displayed the CFS/CNO group spent significantly more time in the target quadrant to search for the hidden platform compared with the CFS/Saline mice (Figure 7i-k). Consist with the improved spatial learning and memory in CFS/CNO mice, NeuN staining revealed that the numbers of surviving CA1 neurons were robustly and significantly elevated (Figure 7l, m). These new results confirmed that chemogenetic activation of the *Th*:LC-CA1 pathway also rescues CFS-induced hippocampal vulnerability to TGI.

3. The statistics are not described in the Methods, and in crucial areas (such as in Figs 2 and 3) appear to be T tests between a control group and depression group in TGI. However, one cannot do a T test in a study with >2 groups, and the testing in this manuscript does not appear to be appropriate.

Response: Thanks for the reviewer's important and valuable comments. In the current study, all statistical analyses were performed using the Graphpad Prism software. We performed the unpaired two-tailed Student's t-test for the experiments with only 2 groups, one-way analysis of variance (ANOVA) with Fisher's LSD post hoc test for the single factor experiments with >2 groups, two-way ANOVA followed by multiple comparisons with Fisher's LSD post hoc test for the double factor experiments. In the water maze test, we performed the repeated-measures (RM) ANOVA for the training phase (6 days), and ordinary two-way ANOVA followed by multiple comparisons with Fisher's LSD post-hoc test for the 6th-day escape latency, length, and probe test. All the detailed P values of the ANOVA parameters (row, column factor and interaction) and multiple comparisons have been reported in corresponding Figure legends (with highlights) in revised manuscript.

4. The quantification Method for the Th+ neuronal arbors from the LC is not described—just that these arbors were imaged and then aligned with an Allen Brain atlas.

Response: Thanks for reviewer's valuable comments. The detailed quantification method for the TH⁺ neuronal arbors from the LC in the Materials and Methods part of manuscript has been described in depth revising.

5. In the DREADD studies, it is now well recognized that CNO is metabolized to clozapine, which directly acts on the dopamine system—which is the system under study in this manuscript. The authors need to comment on the specificity of their work, and how it established that the gain and loss of function are due to DREADD GPCR activation, and not clozapine-dopamine receptor blockade.

Response: Thanks for the reviewer's meaningful comments. In our DREADD studies, we have fully considered the specificity of CNO on the dopamine system. Firstly, we microinjected CNO directly into CA1 region but was not injected via intraperitoneal injection (i.p.). This kind of method greatly limited the effect of CNO metabolite clozapine on dopamine system. Secondly, the results of electrophysiological recording by using MED64 (Supplementary Figure 3a-d) demonstrated that CNO did not alter the spiking frequency of LC neurons with mCherry-overexpression (CNO/mCherry group) (Supplementary Figure 3e, f).

However, CNO incubation significantly reduced or increased the spiking frequency of LC neurons with hMD4i- or hMD3q-overexpression (CNO/hM4Di or CNO/hM3Dq group respectively) (Supplementary Figure 3e, g, h). Therefore, the inhibition or activation of the *Th*: LC-CA1 circuit is associated with the specific function of DREADD system but not an effect of clozapine.

In accordance with the electrophysiological results (Supplementary Figure 3), our gain and loss of function studies also support the specific role of the DREADD system but not an effect of clozapine-dopamine receptor blockade (Figure 5 and Figure 6). In Figure 5, CNO injection did not affect cognition and CA1 neuronal survival in mCherry/TGI-10min group compared to mCherry/Sham group, indicating that clozapine function did not aggravate hippocampal injury upon TGI-10min. Thus, the aggravated hippocampal injury in hM4Di/TGI-10min group after CNO injection is a specific effect of DREADD GPCR inhibition (Figure 5). In Figure 6, CNO injection in mCherry/TGI-10min group showed evident cognition impairment and CA1 neuronal loss after CSDS compared to mCherry/Sham group, suggesting that clozapine function did not affect our modeling of CSDS-induced TGI-10min hippocampal injury. Therefore, we consider that the rescue effect of CNO injection in hM3Dq/TGI-10min group is due to a specific effect of DREADD GPCR activation.

Minor Concerns

Though the authors state at the end of the introduction that TGI-10min was used for all studies, the alternate labeling of TGI (Fig 2-3) vs TGI-10 (Fig 5-6) is potentially confusing to the reader.

Response: We much appreciate and agree with this important suggestion. The labeling of TGI-10min was revised throughout the manuscript and figures to avoid the potentially confusing.

Response to Reviewer comment # 2:

Reviewer #2 (Remarks to the Author):

In this study, Zhang et al., show that depression in mice, caused by chronic social defeat stress (CSDS) or chronic footshock stress (CFS), induced an elevated response to TGI that caused memory impairment and was found to be mediated by the *Th*:LC-CA1 pathway. In addition, they show that selective chemogenetic inhibition of TH+ projections from the LC to the CA1 mimics depression-like behaviors and

induces a worsening response to TGI. Furthermore, the authors claim that chemogenetic activation of the Th:LC-CA1 pathway can reverse these aggravated TGI responses after CSDS. This is a very interesting and laborious study that provides novel insights into the mechanisms of the interplay between depression and TGI and how this affects hippocampal-dependent memory. However, there are also several aspects of the work that are problematic and are presently unconvincing. Validation and control experiments are warranted to strengthen the interpretation of the overall hypothesis.

Response: We greatly appreciate the reviewer for the positive comments on the novelty and workload of this work. We have carefully addressed all the reviewer's concerns and have performed all required experiments to strengthen the interpretation of the overall hypothesis.

1. It is not clear how statistical analysis of the water maze training data (6 days) is performed. The authors mention a two-way ANOVA analysis. However, a repeated-measures ANOVA should be performed. In addition, statistical values need to be reported. Is RM-ANOVA significant and for what parameters? In case of significance, which post-hoc analysis is performed? By looking at the graphs, in most cases there are differences on day 6, but I am not sure that this is enough to justify significance in RM-ANOVA. The authors should report all statistical values better and do that analysis more carefully.

Response: Thanks for the reviewer's valuable and detailed comments. We have revised the statistical analysis section of Materials and Methods part in this manuscript. In the water maze test, we performed the repeated-measures (RM) ANOVA for the training phase (6 days), and ordinary two-way ANOVA followed by multiple comparisons with Fisher's LSD post-hoc test for the 6th-day escape latency, length, and probe test. All the detailed P values of the ANOVA parameters (row, column factor and interaction) and multiple comparisons have been reported in corresponding Figure legends (with highlights) in revised manuscript.

2. Following the chronic social defeat stress paradigm, do the authors observe comparable proportions of susceptible and resilient mice as described in previous reports (e.g. Golden et al, 2011, Nat Protoc)? If not, this should be explained. Additionally, the authors write, "As expected, susceptible mice showed marked decreases in social avoidance in the presence of CDI mice", which is in conflict with the displayed data. The correct phrasing of "decreased social interaction" or "increased social avoidance" should be used.

Response: Thanks for the reviewer's meaningful comments. As the previous

reports (Golden et al, 2011, Nat Protoc)¹, the proportions of susceptible and resilient mice were 62.4 % and 37.6 % separately after CSDS treatment. In Fig. 2, 4, 6 and Supplementary Figure 5 of our study, approximately 60 % mice show a susceptible phenotype and 40 % individuals display a resilient phenotype. Therefore, the proportions of susceptible and resilient phenotype in present study were similar with the previous reports (Golden et al, 2011, Nat Protoc)¹.

Meanwhile, the sentence of “As expected, susceptible mice showed marked decreases in social avoidance in the presence of CD1 mice” have been revised as “As expected, susceptible mice showed markedly decreased social interaction in the presence of CD1 mice” according to the reviewer’s significant suggestion.

3. The authors use a hChR2-YFP fusion protein to label axons originating from the LC in the CA1 region. However, the axonal labeling in their high magnification panels (Fig. 4f) does not look convincing. More needs to be done to prove this signal truly represents LC terminals. To be more convincing, they should co-stain these sections with an anti-TH antibody and show colocalization, and then quantify co-staining with TH to know only LC terminals are stained. Moreover, if their claim is true, a reduction in overall TH signal should be apparent in their susceptible brains (given that TH⁺ axons from the VTA should be much less abundant than those from the LC axons).

Response: Thanks for the reviewer’s significant comments. We have therefore performed a series of supplementary experiments that can improve to demonstrate the axonal labeling from the LC-TH⁺ neurons in the CA1 region. Firstly, the axonal labeling in high magnification panels were revised and showed as new Figure 4f by employing confocal fluorescence microscopy. Simultaneously, an anti-TH antibody was used for co-staining with the CA1 region-including brain sections (Figure 4f). Then, the relative EYFP⁺ pixels density of CA1 area (Figure 4g), and the proportion of EYFP⁺/TH⁺ signal in the whole TH⁺ population (Figure 4h) was conducted. In accordance with the result of EYFP⁺ signal in CA1 region (Figure 4g), the overall TH⁺ signal of susceptible group was apparently decreased when compare with the control and resilient mice (Figure 4i). Moreover, the variation of the *Th*: LC–CA1 pathway was also verified in the CFS mice model (Supplementary Figure 2a-g). Consistent with the reduced *Th*: LC-CA1 projection in susceptible mice after CSDS treatment, the *Th*: LC-CA1 circuit showed a similar decline, containing the proportion of EYFP⁺ signal (Supplementary Figure 2e) and the overall TH⁺ signal (Supplementary Figure 2g) in the CA1 region during the mice subjected to CSF treatment.

4. Their chemogenetic activation data are also problematic. In the water maze probe trial, the saline/susceptible group shows no significant memory deficits compared with the resilient groups. This result is a major issue and does not agree with the

initial hypothesis. Although CNO treatment clearly shows memory improvement, it looks like it is also significantly different from the resilient group. This difference is mainly due to the poor performance of all other groups (about 25% of time spent in target quadrant, something you see also in Fig. 2k). Again, the statistical details should be provided, and comparisons need to be made between all groups.

Response: Thanks for the reviewer's significant comments and pointing these out. We are sorry for not showing correctly and stating clearly of the water maze probe trail data in all Figures, especially of chemogenetic activation result in Figure 6 and the percentage of time spent in target quadrant in Figure 2k. We have carefully double checked the original track and the data of the water maze probe test, the poor performance of the memory test is mainly caused by the incautious labelling the quadrants which caused the off-centered location of the removed platform in the target quadrant during the track analysis. Therefore, we re-performed the probe trail analysis using the original track data with the video-tracking software. The new panels were displayed in the current revision.

Additionally, all the statistical details between all groups of figures were provided in the corresponding figure legends according to the reviewer's helpful suggestion.

5. A control experiment is absent in Figures 5 and 6. CNO should be injected in CA1 of animals that received LC infection of mCherry-only expressing virus. This is necessary to demonstrate specificity of CNO administration on DREADD activation or inhibition of LC terminals. Moreover, functional expression of the DREADDs is not demonstrated using either electrophysiological recording of LC neurons with bath application of CNO or with microdialysis (or comparable sampling methodology) to assay suppression or enhancement of NE/DA release in CA1 with CNO application.

Response: We much appreciate and agree with these important suggestions. Firstly, the control experiments including the CNO/mCherry group were conducted and displayed as the revised Figure 5 and Figure 6 in the revised manuscript. The result suggested that the CNO/mCherry group has no significant effect on the performance related to spatial learning and memory in revised Figure 5d-j of chemogenetic suppression, and Figure 6d-j of chemogenetic enhancement separately. In accordance with the MWM test, the CNO/mCherry mice showed no CA1 neuronal death in revised Figure 5k-l and Figure 6k-l. All the above experiments revealed only CNO treatment, in absence of exogenous DREADDs GPCR expression, was not responding to depressive stress-induced hippocampal vulnerability to transient global ischaemia in the chemogenetic manipulation studies.

Moreover, functional experiment of the DREADDs, containing mCherry-only, hM4Di, and hM3Dq group, was demonstrated by using MED64-based electrophysiological recording. Briefly, the spiking of LC neurons was recorded by multi-electrode array with bath application of CNO *in vitro*. The

electrophysiological recording data suggested the LC neurons was suppression or enhancement in hM4Di and hM3Dq group after CNO stimuli. Additionally, the spiking of LC neurons in mCherry-only group showed no significant response (Supplementary Figure 3a-h).

6. Why do the authors use Nissl staining in Figure 3, while they use NeuN immunoreactivity in all other instances? They have to show quantification of NeuN in Figure 3 as well.

Response: Thanks for the reviewer's helpful comments. The Nissl staining result in Figure 3j-k has been replaced with anti-NeuN immunofluorescence staining data. Meanwhile, the numbers of NeuN⁺ neurons within CA1 region were quantified from the immunofluorescence data, and then the new Figure 3j-k was displayed in the revised manuscript.

7. Panels of Figure 2c-d are mislabeled in the text as 1c-d. Also in Figure 2d target/no target labels should be inverted

Response: Thanks for the reviewer's careful review. All the mistakes about Fig. 1 and 2 have been checked and revised.

More Minor issues:

6. A non-trivial percentage of LC neurons which were TH-negative expressed mCherry after viral infection (~10%). Can the authors comment on this population? Were the projection targets of the mCherry+ TH- LC neurons identified? What is the neurotransmitter synthesized/released from these neurons?

Response: Thanks for reviewer's valuable comments. The reasons why the approximately 10 % of LC neurons which were TH-negative expressed mCherry after viral infection maybe attribute to methodological problems about immunofluorescence staining as following. Firstly, in order to avoid the fluorescence quenching of mCherry-positive signal, the brain frozen section was pretreatment with 1% sodium dodecyl sulfate (SDS) for antigen retrieval². The SDS-based method for antigen retrieval was more moderate than other methods such as heating and Citrate Buffer (10 mM Sodium Citrate Buffer, 0.05 % Tween 20, pH 6.0)^{3, 4}. Therefore, the TH antigen was not absolutely retrieval for subsequent immunofluorescence staining. Secondly, the antigen/antibody combination efficiency of polyclonal anti-TH antibody was not full in the experimental process of immunofluorescence staining. This problem maybe also leads to ~10% of LC neurons were TH-negative expressed mCherry after viral infection. Collectively, the identified mCherry⁺TH⁻ population in our study were actually LC-TH⁺ neurons. Previous findings provide direct evidence that

projections from the LC are not purely noradrenergic, but co-release dopamine in the dorsal CA1 of hippocampus⁵.

7. Resilient mice appear to demonstrate diminished LC-derived TH staining in LHA, ML, and SNc. What does this mean? Is neuronal survival negatively impacted in these regions after the chronic stress and TGI as observed in the susceptible mice after CSDS?

Response: Thanks for the reviewer's significant comments. The additional experiments were performed to clarify whether the neuronal survival negatively impacted in these regions (LHA, ML, and SNc) after the chronic stress and TGI as observed in the susceptible mice after CSDS treatment. The neuronal survival of LHA, ML, and SNc region by anti-NeuN⁺ immunofluorescence (Supplementary Figure 1a). The results suggested that the neuronal survival was no significant difference among the control, susceptible, and susceptible/TGI-10min groups in LHA (Supplementary Figure 1b), ML (Supplementary Figure 1c), and SNc (Supplementary Figure 1d) region.

8. The methodology is well outlined, but there are numerous grammatical errors. This section, in particular, requires further proof-reading and editing by the authors.

Response: Thanks for the reviewer's meaningful comments. The manuscript has been polished by professional and native English language editors in depth editing. Meanwhile, the grammatical errors of Materials and Methods section have been corrected in revised manuscript.

References

1. Golden SA, Covington HE, 3rd, Berton O, Russo SJ. A standardized protocol for repeated social defeat stress in mice. *Nat Protoc* **6**, 1183-1191 (2011).
2. Brown D, Lydon J, McLaughlin M, Stuart-Tilley A, Tyszkowski R, Alper S. Antigen retrieval in cryostat tissue sections and cultured cells by treatment with sodium dodecyl sulfate (SDS). *Histochem Cell Biol* **105**, 261-267 (1996).
3. Ino H. Antigen retrieval by heating en bloc for pre-fixed frozen material. *J Histochem Cytochem* **51**, 995-1003 (2003).
4. Evers P, Uylings HB. Effects of microwave pretreatment on immunocytochemical staining of vibratome sections and tissue blocks of human cerebral cortex stored in formaldehyde fixative for long periods. *J Neurosci Methods* **55**, 163-172 (1994).

5. Kempadoo KA, Mosharov EV, Choi SJ, Sulzer D, Kandel ER. Dopamine release from the locus coeruleus to the dorsal hippocampus promotes spatial learning and memory. *Proc Natl Acad Sci U S A* **113**, 14835-14840 (2016).

Reviewers' comments:

Reviewer #1 (Remarks to the Author):

This manuscript has been modified to address this Reviewer's original concerns. Substantial new experiments have been added, with new data and figures.

Reviewer #2 (Remarks to the Author):

The revised manuscript is significantly improved from its initial version. The authors included a significant amount of new data and important control experiments that strengthen the paper considerably. They have also included statistical analyses that were lacking in the previous version. These analyses are an important addition, but there are a number of problems with their statistics, which will need to be visited again.

1. The authors use one-way and repeated measures ANOVA correctly, but for their post-hoc analysis they utilize the Fisher LSD method. This should not be the method of choice because the Fisher does not take into account multiple comparisons. Instead, they should use alternative post-hoc analyses like Tukey, Dunnet, or Bonferroni tests in which they adjust the p value to account for multiple comparisons. Their statistical software of choice (Graphpad Prism) offers great suggestions for the appropriate post-hoc test to be used in each case and they should re-analyze their data using those suggestions.

2. Moreover, the authors must revisit their symbols for significance. Using multiple symbols (*, #, &...) without proper explanations is very confusing. Instead of this, they should use only stars and include which parameters are significant. Doing that they should also be careful to report the p values correctly and according to what they describe in their methodology section. There are many instances that p values larger than 0.001 are reported with 3 stars instead of two (see lines 974, 975, 985).

3. Their water maze data in Fig 1f and 1h suggest that there are no significant differences over time between their groups (the RM ANOVA for treatment or interaction is not significant). The only significant parameter is time which denotes that the test works (all mice reduce their escape latency over time). This result was expected from the graph that also shows that the significant differences occur after day 6. This finding should be made clear by the authors because the appearance of significance in the figure comes only by the effect of time, and it is not properly presented.

4. Line 536 "...and dehydrated with the gradient of sucrose...". Usually, sucrose gradients are used for cryoprotection and not for dehydration and typically disrupt interactions between polar water molecules.

Point-by-point response to the referees' comments

Response to Reviewer comment # 1:

Reviewer #1 (Remarks to the Author):

This manuscript has been modified to address this Reviewer's original concerns. Substantial new experiments have been added, with new data and figures.

Response: We greatly appreciate the reviewer for the positive comments on the novelty and workload of this work.

Response to Reviewer comment # 2:

Reviewer #2 (Remarks to the Author):

The revised manuscript is significantly improved from its initial version. The authors included a significant amount of new data and important control experiments that strengthen the paper considerably. They have also included statistical analyses that were lacking in the previous version. These analyses are an important addition, but there are a number of problems with their statistics, which will need to be visited again.

Response: We thank this reviewer for the positive review and constructive comments on our manuscript. According to these significant comments, we have revised the manuscript as follows.

1. The authors use one-way and repeated measures ANOVA correctly, but for their post-hoc analysis they utilize the Fisher LSD method. This should not be the method of choice because the Fisher does not take into account multiple comparisons. Instead, they should use alternative post-hoc analyses like Tukey, Dunnett, or Bonferroni tests in which they adjust the p value to account for multiple comparisons. Their statistical software of choice (Graphpad Prism) offers great suggestions for the appropriate post-hoc test to be used in each case and they should re-analyze their data using those suggestions.

Response: We greatly appreciate the reviewer for the significant suggestions. We have re-done the post hoc analysis using Dunnett's test instead of Fisher LSD test according the reviewer's comments. The statistical results were reported in the revised figures and figure legends.

2. Moreover, the authors must revisit their symbols for significance. Using multiple symbols (*, #, &...) without proper explanations is very confusing. Instead of this,

they should use only stars and include which parameters are significant. Doing that they should also be careful to report the p values correctly and according to what they describe in their methodology section. There are many instances that p values larger than 0.001 are reported with 3 stars instead of two (see lines 974, 975, 985).

Response: Thanks for the reviewer's meaningful comments. We have remarked the symbols for significance in the two-way ANOVA using only stars instead of multiple symbols (*, #, &...), and gave clear indication which parameters are significant in the figures. The p values were carefully reported in the figure legends according to the methodology. Moreover, the number of stars were checked and revised in all figures.

3. Their water maze data in Fig 1f and 1h suggest that there are no significant differences over time between their groups (the RM ANOVA for treatment or interaction is not significant). The only significant parameter is time which denotes that the test works (all mice reduce their escape latency over time). This result was expected from the graph that also shows that the significant differences occur after day 6. This finding should be made clear by the authors because the appearance of significance in the figure comes only by the effect of time, and it is not properly presented.

Response: We are grateful to the reviewer for the valuable suggestions that made our manuscript markedly improved. The aim of experiments in Fig. 1 was to identify the time axis of spatial learning/memory impairment and hippocampal neuron death caused by transient global ischemia (TGI) at different times, therefore determine the appropriate time-point of TGI treatment for the following depression-induced hippocampal vulnerability to transient global ischaemia experiments.

As the reviewer pointed out, the results of the six-day learning curve in Fig. 1f and 1h showed that there was no significant difference in spatial learning ability (escape latency and swimming length) between the different treatment groups. However, the escape latency (Fig. 1g) and swimming length (Fig. 1i) of the 6th day were significantly lengthened in TGI-20min and TGI-30min, but not TGI-10min group, compared with sham-operated mice. Moreover, the time in target quadrant in the 7th-day probe trail (Fig. 1j and 1k) and the survival of CA1 neurons (Fig. 1m and 1n) were also significantly decreased in TGI-20min and TGI-30min, but not TGI-10min, compared with Sham-operated group.

The above experimental results show that TGI for more than 20 minutes could cause the impairment of learning (at the later stage of training session) and memory, and also loss of the CA1 neurons, while 10-min TGI had no obvious damage to mice. Thus, we chose 10 minutes as the optimal time point of TGI

treatment for the following depression-induced hippocampal vulnerability to transient global ischaemia experiments.

4. Line 536 “...and dehydrated with the gradient of sucrose...”. Usually, sucrose gradients are used for cryoprotection and not for dehydration and typically disrupt interactions between polar water molecules.

Response: We much appreciate and agree with this important suggestion. The sentence of “...and dehydrated with the gradient of sucrose...” was corrected to “...and treated with the gradient of sucrose for cryoprotection...”.

REVIEWERS' COMMENTS:

Reviewer #2 (Remarks to the Author):

These findings seem strong, and the response of the authors was adequate.

Point-by-point response to the reviewers

Response to REVIEWERS' COMMENTS:

Reviewer #2 (Remarks to the Author):

These findings seem strong, and the response of the authors was adequate.

Response: We greatly appreciate the reviewer for the positive comments of this work.